# Extensive remodelling of the cell wall during the development of *Staphylococcus aureus* bacteraemia

Edward JA Douglas[1,2†], Nathanael Palk[1†], Tarcisio Brignoli[1,3], Dina Altwiley[1], Marcia Boura[1], Maisem Laabei[2], Mario Recker[4,5], Gordon YC Cheung[6], Ryan Liu[6], Roger C Hsieh[6], Michael Otto[6], Eoin O'Brien[7], Rachel M McLoughlin[7], Ruth C Massey[1,8*]

[1]School of Cellular and Molecular Medicine, University of Bristol, Bristol, United Kingdom; [2]Department of Life Sciences, University of Bath, Bath, United Kingdom; [3]Department of Biosciences, Università degli Studi di Milano, Milan, Italy; [4]Institute of Tropical Medicine, University of Tübingen, Tübingen, Germany; [5]Centre for Ecology and Conservation, University of Exeter, Penryn Campus, Exeter, United Kingdom; [6]Laboratory of Bacteriology, National Institute of Allergy and Infectious Diseases (NIAID), US National Institutes of Health (NIH), Bethesda, United States; [7]Host Pathogen Interactions Group, School of Biochemistry and Immunology, Trinity College Dublin, Dublin, Ireland; [8]Schools of Microbiology and Medicine, University College Cork, and APC Microbiome Ireland, Cork, Ireland

*For correspondence:
ruth.massey@bristol.ac.uk

†These authors contributed equally to this work

Competing interest: The authors declare that no competing interests exist.

**Abstract** The bloodstream represents a hostile environment that bacteria must overcome to cause bacteraemia. To understand how the major human pathogen *Staphylococcus aureus* manages this we have utilised a functional genomics approach to identify a number of new loci that affect the ability of the bacteria to survive exposure to serum, the critical first step in the development of bacteraemia. The expression of one of these genes, *tcaA*, was found to be induced upon exposure to serum, and we show that it is involved in the elaboration of a critical virulence factor, the wall teichoic acids (WTA), within the cell envelope. The activity of the TcaA protein alters the sensitivity of the bacteria to cell wall attacking agents, including antimicrobial peptides, human defence fatty acids, and several antibiotics. This protein also affects the autolytic activity and lysostaphin sensitivity of the bacteria, suggesting that in addition to changing WTA abundance in the cell envelope, it also plays a role in peptidoglycan crosslinking. With TcaA rendering the bacteria more susceptible to serum killing, while simultaneously increasing the abundance of WTA in the cell envelope, it was unclear what effect this protein may have during infection. To explore this, we examined human data and performed murine experimental infections. Collectively, our data suggests that whilst mutations in *tcaA* are selected for during bacteraemia, this protein positively contributes to the virulence of *S. aureus* through its involvement in altering the cell wall architecture of the bacteria, a process that appears to play a key role in the development of bacteraemia.

## eLife assessment

This **important** study uses an innovative GWAS approach and targeted testing to highlight *S. aureus* genes that modify susceptibility to serum, serum-derived antimicrobial products, and commonly used antibiotics. These findings are significant in that they highlight evidence of evolution of virulence determinants in the setting of exposure to host stressors expected to be present during

bacteremia and antibiotic therapy. **Compelling** results build on a foundation of work attributing loss-of-function mutations in tcaA to glycopeptide non-susceptibility.

## Introduction

*Staphylococcus aureus* is an important human pathogen and a significant global health concern (*Gordon and Lowy, 2008*; *Lowy, 1998*). The most common interaction with its human host is as an asymptomatic coloniser; however, it frequently transitions to a pathogenic state with the ability to cause a wide range of diseases, ranging from relatively minor skin and soft tissue infections (SSTI) to more life-threatening incidents of endocarditis or bacteraemia (*Gordon and Lowy, 2008*; *Lowy, 1998*). *S. aureus* is notorious for producing a plethora of virulence factors ranging from pore-forming toxins to various immune evasion strategies (*Powers and Bubeck Wardenburg, 2014*). The toxicity of *S. aureus* is generally accepted as playing an important role during infection, with high toxicity isolates typically causing more severe symptoms and disease progression (*Abdelnour et al., 1993*; *Cheung et al., 1994*). However, for the more invasive diseases such as bacteraemia and pneumonia, it has been shown that the causative isolates are often impaired in their toxin production and instead rely on alternative virulence approaches to cause disease, such as being able to better survive exposure to host immune defences (*Rose et al., 2015*; *Laabei et al., 2015*).

As the most severe type of infection caused by *S. aureus*, bacteraemia has a mortality rate of between 20% and 30% (*Kaasch et al., 2014*; *Bai et al., 2022*), and despite infection control measures that have reduced the incidence of MRSA bacteraemia, the incidence of MSSA bacteraemia continues to increase in several countries, suggesting we have much to learn about this disease process (*OGL Cookies on GOV.UK, 2023*). Entry into the bloodstream is the first step in the development of bacteraemia and represents a major bottleneck for the bacteria, whether they seed from an infection elsewhere in the body, or take a more direct route through an intravenous device. As a heavily protected niche, the bloodstream contains numerous humoral immune features with potent anti-staphylococcal activity, including antimicrobial peptides (AMPs) and host defence fatty acids (HDFAs) (*Levy, 2000*; *Kenny et al., 2009*; *Beavers et al., 2019*; *Das, 2018*). However, the fact that cases of *S. aureus* bacteraemia occur demonstrates that these do not represent an impregnable force and that the bacteria can adapt to resist these defensive features.

In previous work we have pioneered the application of genome-wide association studies (GWAS) to characterise bacterial virulence, where it has proven to be a powerful approach to define complex regulatory pathways (*Laabei et al., 2015*; *Laabei et al., 2014*; *Recker et al., 2017*; *Yokoyama et al., 2018*; *Stevens et al., 2022*; *Altwiley et al., 2021*). In related work we have also begun to characterise a new category of *S. aureus* genes we refer to as MALs (mortality associated loci) and found that many of these are involved in serum survival (*Douglas et al., 2021*), with serum containing the bulk of the humoral immune response present in blood, but without the added complexity of the cellular components. Given the outstanding research questions surrounding the development of bacteraemia, we performed a GWAS on a collection of 300 clinical isolates with respect to serum resistance. We found significant variability in how well individual isolates survive exposure to serum and identified seven novel effectors of this activity. Of particular note was the TcaA protein, which we have further characterised and demonstrate that it is a critical protein for the bacteria involved in the remodelling of the bacterial cell wall during the development of bacteraemia.

## Results

### Survival of *S. aureus* upon exposure to human serum is a variable trait

The human bloodstream is a highly protected niche, and so to establish an infection in this environment the bacteria must evade many aspects of host immunity, such as the bacterial membrane damaging AMPs and HDFAs found in serum (*Levy, 2000*; *Kenny et al., 2009*; *Beavers et al., 2019*; *Peschel et al., 1999*; *Ernst and Peschel, 2011*; *Joo and Otto, 2015*; *Kohler et al., 2009*). To examine the level of variability that exists in serum susceptibility in natural populations of *S. aureus*, we exposed 300 clinical isolates to pooled human serum and quantified the proportion of each culture that survived. These 300 isolates represent the two major clones, as defined by multi-locus sequence typing: clonal complex 22 (CC22) and clonal complex 30 (CC30), that are responsible for the majority

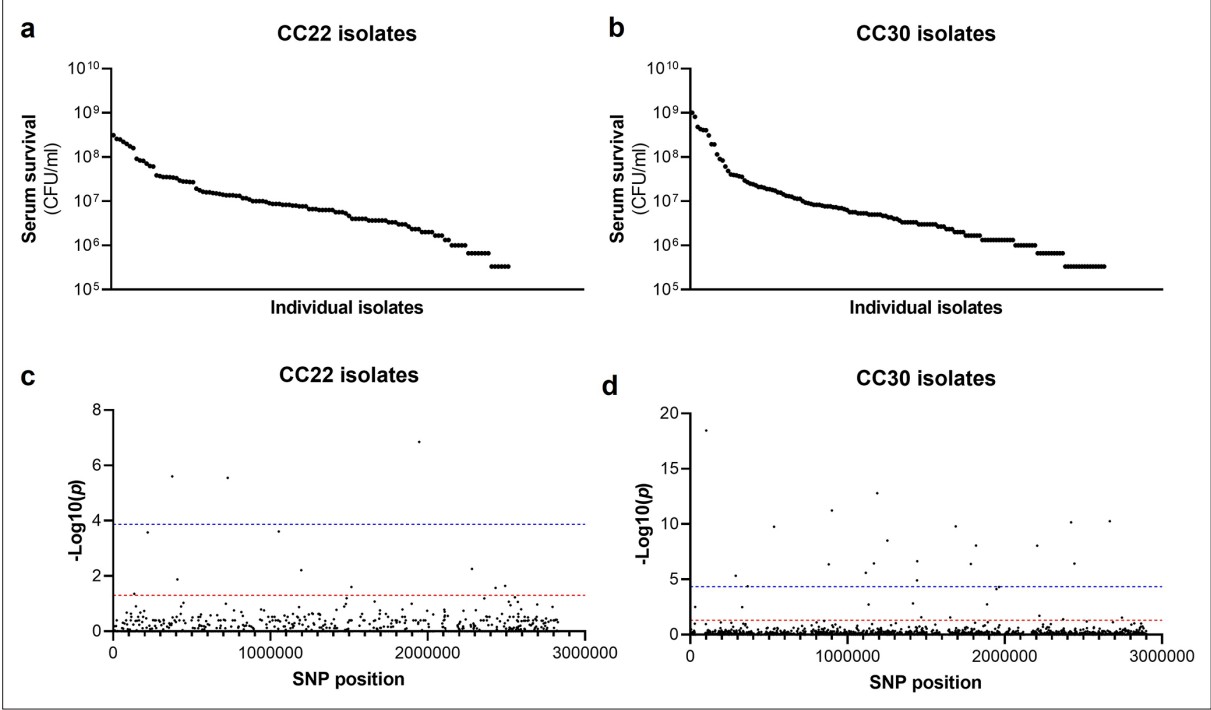

**Figure 1.** Serum susceptibility is multifactorial and varies widely across closely related *S. aureus* isolates. (**a** and **b**) The susceptibility of 300 bacteraemia isolates from clonal complexes clonal complex 22 (CC22) and CC30 were exposed to human serum and their ability to survive this exposure quantified. The survival of each isolate was quantified in triplicate and dot represents the mean of these data. (**c** and **d**) Manhattan plots representing the statistical associations (on the y axis) between individual single nucleotide polymorphisms (SNPs) across the genome (on the x axis) and serum survival. The dotted red line represents the uncorrected significance threshold, and the dotted blue line represents the Sidak corrected (for multiple comparisons) threshold.

of MRSA cases in the UK and Ireland (*Recker et al., 2017*). The isolates were collected from 300 individual cases of bacteraemia. This collection is a unique resource as each strain has been sequenced, extensively phenotyped, and we have clinical metadata for many of the patients the bacteria were isolated from *Recker et al., 2017*. We found there to be similar levels of variability in the ability of the bacteria to survive exposure to human serum across the two CCs, both with a >$10^3$-fold difference between the most and least susceptible isolate with respect to the number of bacteria that survived (*Figure 1a and b*).

## Survival of *S. aureus* upon exposure to human serum is a polygenic trait

As the genome sequence for each of the 300 clinical *S. aureus* isolates was available, we performed a GWAS to identify polymorphic loci or single nucleotide polymorphisms (SNPs) that associated with the level of susceptibility of isolates to serum. For this, the data from the two distinct clones were analysed independently, with population structure within the clones being accounted for (*Figure 1c and d*, *Table 1a and b*). We applied both uncorrected and corrected (for multiple comparisons) significance thresholds to this analysis, as our previous work has demonstrated that the stringency of multiple correction approaches increases the likelihood of type II errors (i.e., false negative results). For the CC22 collection 12 loci were associated with serum survival and for the CC30's there were 32 associated loci (*Figure 1c and d*, *Table 1a and b*).

## Functional validation of the GWAS results identifies seven novel genes that affect serum survival

Of the 44 loci associated with serum survival, there were transposon mutants available for 32 in the Nebraska library (*Fey et al., 2013*). To functionally verify our GWAS findings, each of these mutants was tested for their ability to survive exposure to human serum. We found that seven mutants were significantly affected in this ability, and this effect was complemented by expressing the inactivated gene in trans (*Figure 2*). The pRMC2 plasmid used to complement the mutated phenotypes requires

**Table 1.** Loci associated with serum survival in the clonal complex 22 (CC22) (a) and CC30 (b) collections.

The single nucleotide polymorphism (SNP) position is relative to the origin of replication in the reference genomes: HO 5096 0412 (CC22) and MRSA252 (CC30). Locus tags and where available gene names or putative protein functions have been provided. NTML refers to the mutant available in the Nebraska transposon library.

**a**

| SNP position | CC22 locus tag | Description | Log p-Value | NTML |
|---|---|---|---|---|
| 1945336 | SAEMRSA15_RS09605 | Intergenic | 6.851546 | N/A |
| 375179 | SAEMRSA15_03110 | metE | 5.606418 | NE944 |
| 728154 | SAEMRSA15_06310 | saeS | 5.553331 | NE1296 |
| 1053036 | N/A | Intergenic | 3.60927 | N/A |
| 219764 | SAEMRSA15_01760 | Putative oxidoreductase | 3.575929 | NE312 |
| 2280500 | SAEMRSA15_20910 | Alcohol dehydrogenase | 2.257841 | NE1147 |
| 1196127 | SAEMRSA15_10750 | yfhO | 2.212486 | NE1597 |
| 408213 | SAEMRSA15_03430 | guaA | 1.876101 | N/A |
| 2492381 | SAEMRSA15_23120 | tcyB | 1.645759 | NE228 |
| 1514441 | SAEMRSA15_13420 | Hypothetical protein | 1.600037 | NE1349 |
| 2430758 | SAEMRSA15_22540 | tcaA | 1.572702 | NE285 |
| 134276 | SAEMRSA15_01100 | Hypothetical protein | 1.358386 | NE599 |

**b**

| SNP position | CC30 locus tag | Description | Log p-Value | NTML |
|---|---|---|---|---|
| 101174 | SAR0097 | Putative DNA binding protein | 18.466 | N/A |
| 1188605 | SAR1143 | arcC | 12.786 | NE1719 |
| 900409 | SAR0856 | Phosphoglycerate mutase | 11.226 | NE1422 |
| 2667418 | SAR2584 | gntR | 10.263 | NE1124 |
| 2421610 | SAR2347 | Putative membrane protein | 10.160 | N/A |
| 1687821 | SAR1615 | aroK | 9.793 | N/A |
| 532944 | SAR0495 | veg | 9.751 | NE1703 |
| 1252982 | SAR1203 | recG | 8.502 | NE1344 |
| 1816815 | SAR1754 | clpX | 8.064 | N/A |
| 2203190 | SAR2143 | ilvC | 8.050 | NE1177 |
| 1443567 | SAR1386 | trpA | 6.647 | NE304 |
| 1167449 | SAR1119 | uvrC | 6.439 | NE1212 |
| 2242306 | SAR2374 | ureC | 6.425 | NE410 |
| 1784580 | SAR1719 | tgt | 6.396 | NE1885 |
| 880082 | SAR0835 | est | 6.354 | NE1122 |
| 1116463 | SAR1070 | pdhD | 5.600 | NE1610 |
| 289265 | SAR0248 | tagB | 5.324 | NE362 |
| 1442353 | SAR1385 | trpB | 4.910 | NE310 |
| 364861 | SAR0319 | NADH flavin oxidoreductase | 4.373 | N/A |
| 1963781 | SAR1877 | menE | 4.315 | N/A |

*Table 1 continued on next page*

*Table 1 continued*

**b**

| SNP position | CC30 locus tag | Description | Log p-Value | NTML |
|---|---|---|---|---|
| 1946567 | SAR1856 | *arsB2* | 4.122 | NE1484 |
| 1416353 | SAR1364 | Iron sulphur cluster protein | 2.829 | NE1910 |
| 1887414 | SAR1810 | *fhs* | 2.741 | NE706 |
| 1133886 | SAR0251 | *tarF* | 2.725 | N/A |
| 30969 | SAR0023 | *sasH* | 2.507 | NE295 |
| 330358 | SAR0284 | *essC* | 2.486 | NE1484 |
| 2220348 | SAR2151 | *tex* | 1.704 | NE1079 |
| 1468392 | SAR1410 | *hipO* | 1.569 | NE1688 |
| 1652613 | SAR1582 | *zwf* | 1.556 | N/A |
| 2744774 | SAR2656 | *nrmA* | 1.549 | NE1223 |
| 2370812 | SAR2287 | *lacR* | 1.382 | NE346 |

the use of two antibiotics, one for selection of the plasmid and one for induction of gene expression (*Corrigan and Foster, 2009*). These antibiotics can often affect bacterial gene expression. As such validation of complementation is based on the comparison between the empty and target gene containing plasmid which can be grown under the same conditions. The seven genes were *tcaA*, a

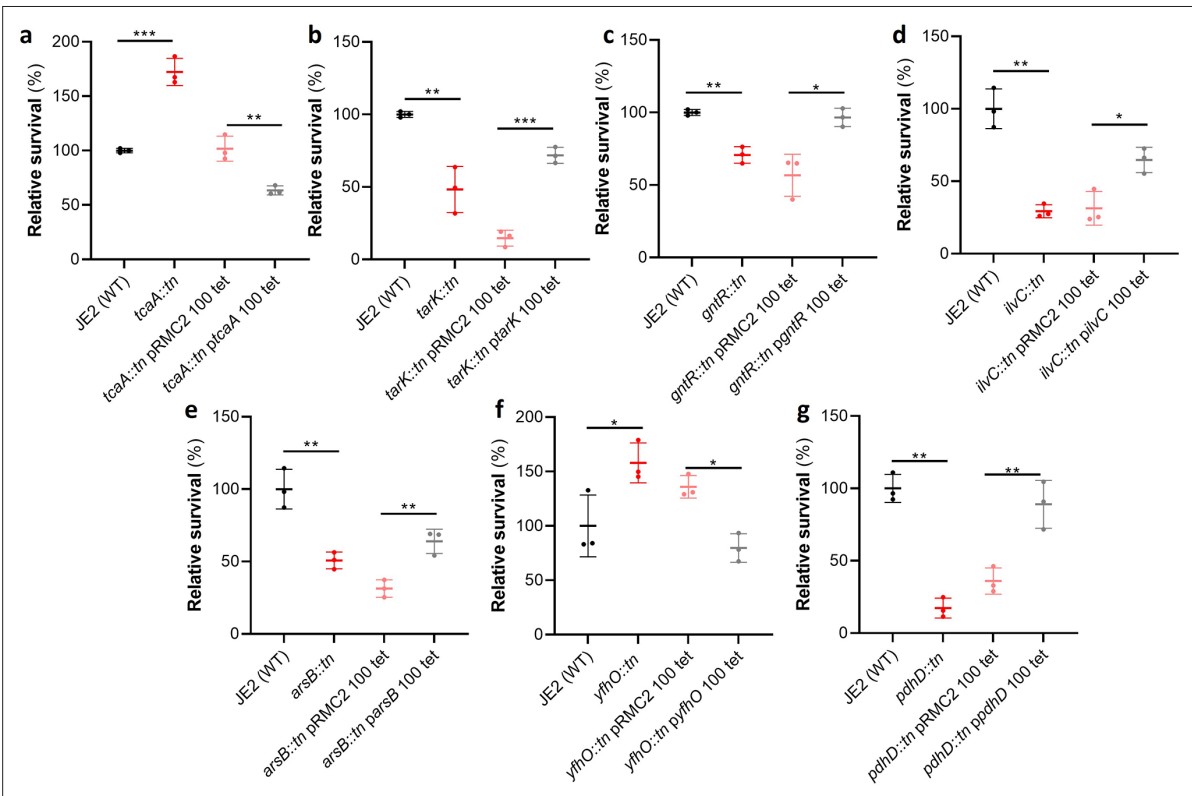

**Figure 2.** Functional verification of loci involved in serum sensitivity of *S. aureus*. The effect of the inactivation of the genes associated serum sensitivity was examined using transposon mutants of strain JE2. Of the 32 mutants tested seven were significantly affected: (**a**) *tcaA*, (**b**) *tarK*, (**c**) *gntK*, (**d**) *ilvC*, (**e**) *arsB*, (**f**) *yhfO*, and (**g**) *pdhD*. The effect of the mutations on serum sensitivity was complemented by expressing the gene from the expression plasmid pRMC2 (e.g., the *tcaA* complementing plasmid is called p*tcaA*). The dots represent individual data points (n=3), the bars the mean value, and the error bars the standard deviation. Significance was determined as *<0.05, **<0.01, ***<0.001.

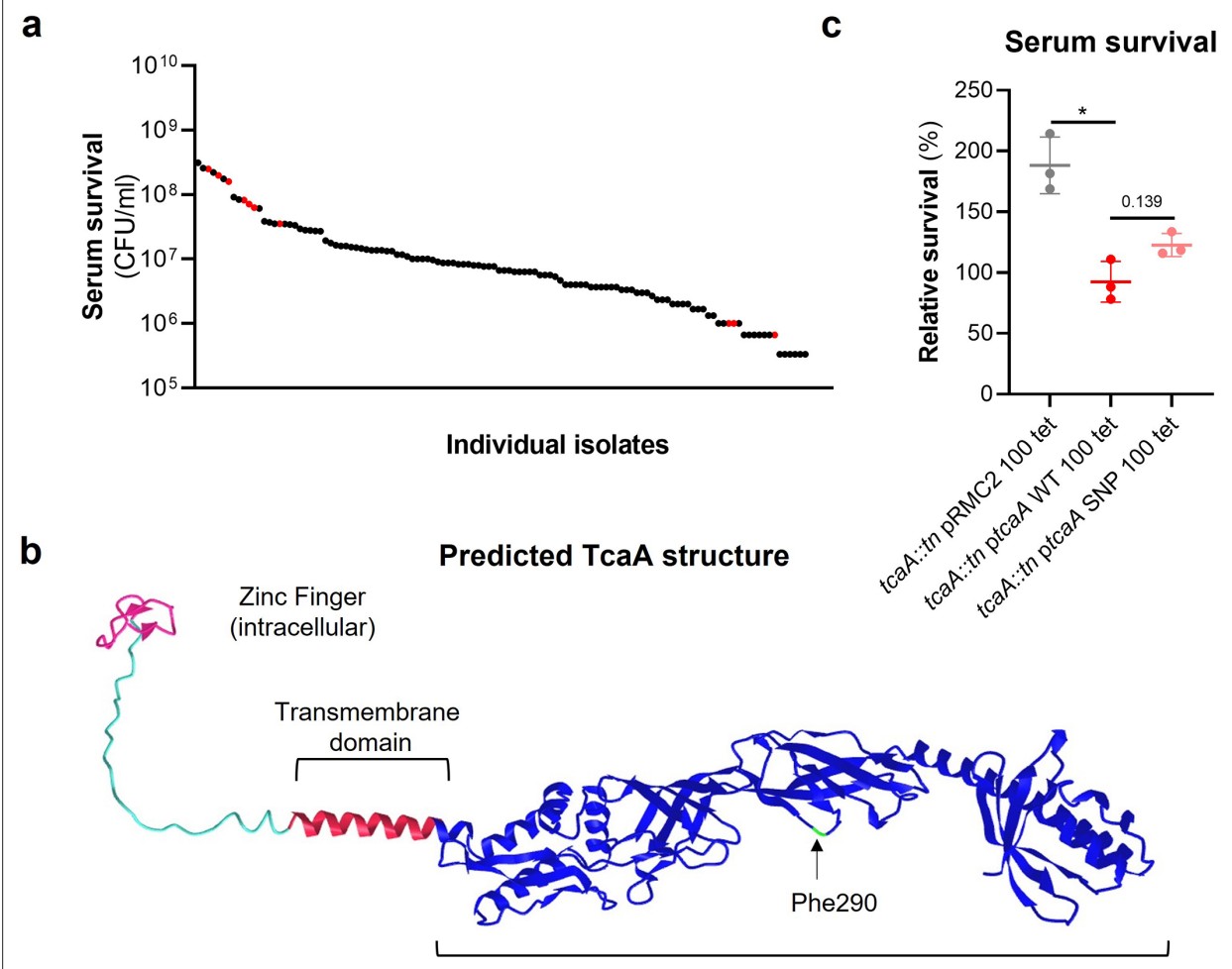

**Figure 3.** *The clinical* tcaA *mutations are associated with increased serum resistance.* (**a**) The individual clinical isolates with polymorphism in the *tcaA* gene are indicated (in red) on a graph displaying the range of serum survival of the collection. (**b**) The structure of the TcaA protein as predicted by AlphaFold. (**c**) The most common *tcaA* single nucleotide polymorphism (SNP) decreases the sensitivity of *S. aureus* to serum. The dots represent individual data points (n=3), the bars the mean value, and the error bars the standard deviation. Significance was determined as *<0.05.

gene associated with resistance to the antibiotic teicoplanin (*Brandenberger et al., 2000*); *tarK*, a gene involved in wall teichoic acid (WTA) biosynthesis (*Xia et al., 2010*); *gntR*, which encodes gluconate kinase (*Fujita and Fujita, 1987*); *ilvC*, which encodes ketol-acid reductoisomerase (*Li et al., 2017*); *arsB*, which is an efflux pump involved in arsenic resistance (*Shen et al., 2013*); *yfhO*, which is involved in the glycosylation of lipoteichoic acid (LTA) (*Rismondo et al., 2018*); and *pdhD*, which encodes a lipoamide dehydrogenase (*Hemilä, 1991*).

## The SNPs in the tcaA gene of the clinical isolates are associated with decreased serum sensitivity

The *tcaA* gene has previously been reported to be involved in conferring increased sensitivity to the antibiotic teicoplanin (*Brandenberger et al., 2000*), although the molecular detail of how it achieves this has yet to be determined, and our findings suggest that it also confers increased sensitivity to serum (*Figure 2a*). To visualise the signal detected by the GWAS for this gene we have mapped the position of the clinical isolates with the *tcaA* SNPs onto a figure displaying the range of serum sensitivities of the collection (*Figure 3a*). The majority of the isolates with the SNPs were less sensitive to serum, suggesting the SNPs negatively affect the activity of the protein. The most common SNP in the *tcaA* gene amongst the clinical isolates conferred a Phe (290) to Ser change in the protein sequence. Using AlphaFold (*Jumper et al., 2021*) we generated a structure for TcaA where it is predicted to be

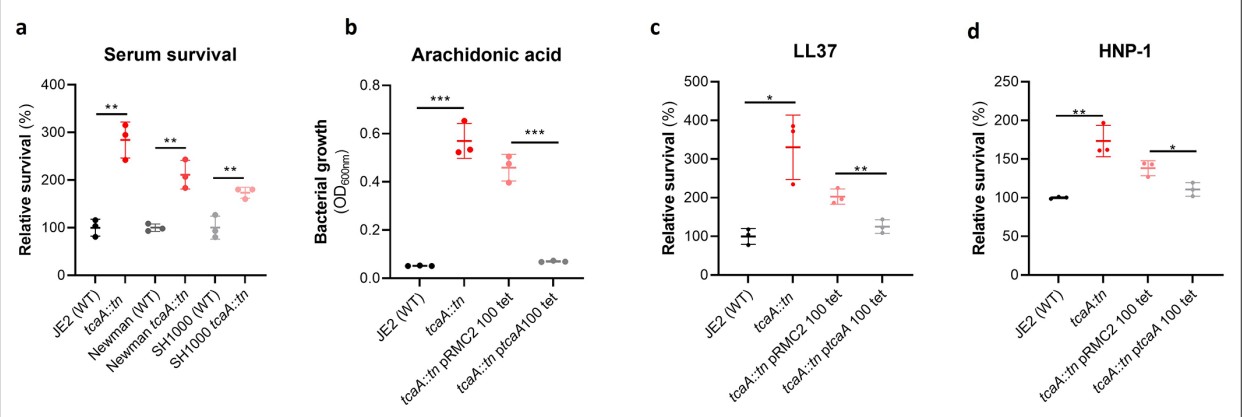

**Figure 4.** TcaA production confers increased sensitivity to several antibacterial components of serum. (**a**) The inactivation of the *tcaA* gene decreases the sensitivity of three *S. aureus* strains (i.e., JE2, Newman and SH1000) to serum (90 min exposure). (**b**) TcaA production by the wild type and p*tcaA* complemented strain confers increased sensitivity to arachidonic acid (100 µM) following overnight growth. (**c**) TcaA production by the JE2 wild type and p*tcaA* complemented strain confers increased sensitivity to LL37 (5 µg/ml, 90 min exposure) and (**d**) TcaA production by the wild type and p*tcaA* complemented strain confers increased sensitivity to HNP-1 (5 µg/ml, 90 min exposure). The dots represent individual data points (n=3), the bars the mean value, and the error bars the standard deviation. Significance was determined as *<0.05, **<0.01, ***<0.001.

The online version of this article includes the following figure supplement(s) for figure 4:

**Figure supplement 1.** TcaA does not contribute to the clumping of *S.aureus* when incubated with serum.

**Figure supplement 2.** Sensitivity of genome-wide association study (GWAS) identified loci to antibacterial components of serum.

membrane bound with a single membrane spanning domain; to have a zinc finger presented intra-cellularly; with the majority of the protein including the Phe290 displayed extracellularly (*Figure 3b*). Given that phenylalanines are frequently involved in protein-protein interactions (*Ma and Nussinov, 2007*) we examined the effect this change has on the activity of TcaA by cloning the *tcaA* gene with this SNP into the complementing pRMC2 plasmid and comparing its ability to confer serum resistance relative to that of the *tcaA* gene without the SNP. The TcaA protein with Ser at position 290 instead of Phe were slightly more resistant to serum killing, but not as affected as when the entire protein was inactivated, suggesting this change has a subtle effect on the serum sensitising activity of the protein (*Figure 3c*).

## TcaA confers increased sensitivity to multiple antibacterial components of serum

To examine whether the effect of TcaA on serum resistance was specific to the JE2 background we transduced the *tcaA::tn* mutation from JE2 into *S. aureus* strains SH1000 and Newman. Attempts were made to transduce this mutation into a clinical CC22 isolate, but this lineage was refractory to any of the available transducing phage (*Waldron and Lindsay, 2006*; *Monk et al., 2012*). In all three backgrounds TcaA increased the sensitivity of the bacteria to serum (*Figure 4a*) verifying that the effect was not limited to the JE2 background. To determine which aspects of human serum TcaA were conferring increased sensitivity to, we measured the relative ability of the wild type and mutant to survive exposure to some of the antibacterial factors found in serum: to the HDFA arachidonic acid, and to two AMPs: HNP-1 and LL37. The strains producing TcaA (i.e., the wild type and complemented strain) were more sensitive to all three components of human serum (*Figure 4b–d*). For the AMPs this sensitivity was significant after only 90 min exposure, whereas for arachidonic acid sensitivity the bacteria were incubated overnight, and differences in sensitivity determined by bacterial growth (OD$_{600}$). Although none of the individual antibacterial components of serum tested here are involved in *S. aureus* cell clumping, it is a possibility that some of the effects seen in serum is as a result of this activity, artificially amplifying the effect. To examine this the wild type and *tcaA* mutant were visually examined by microscopy in both PBS and serum where no differences in clumping was observed (*Figure 4—figure supplement 1*). Of the other GWAS identified genes, the *ilvC* mutant was less sensitive to arachidonic acid, the *yfhO* mutant was more resistant to HNP-1 and LL-37, while the *tarK*,

*gntK*, *arsB*, and *phdD* mutants were all more sensitive to HNP-1 and LL37 when compared to the JE2 wild type strain (*Figure 4—figure supplement 2*).

## WTA contribute to the arachidonic acid sensitivity of TcaA producing strains

The antibacterial properties of HDFAs and AMPs share some common features in that they both penetrate through the cell wall and attack the bacterial membrane (*Levy, 2000*; *Kenny et al., 2009*; *Beavers et al., 2019*; *Peschel et al., 1999*; *Ernst and Peschel, 2011*; *Joo and Otto, 2015*). AMPs are positively charged molecules that rely on the relatively negative charge across the bacterial cell envelope to penetrate, and thus resistance is frequently acquired by changing the charge across the cell wall such that the AMPs are repelled (*Ernst and Peschel, 2011*; *Joo and Otto, 2015*). Less is known about how HDFAs penetrate the cell wall; however, resistance is associated with changes in the abundance of WTA in the bacterial cell walls (*Kohler et al., 2009*). WTA are hydrophilic and when

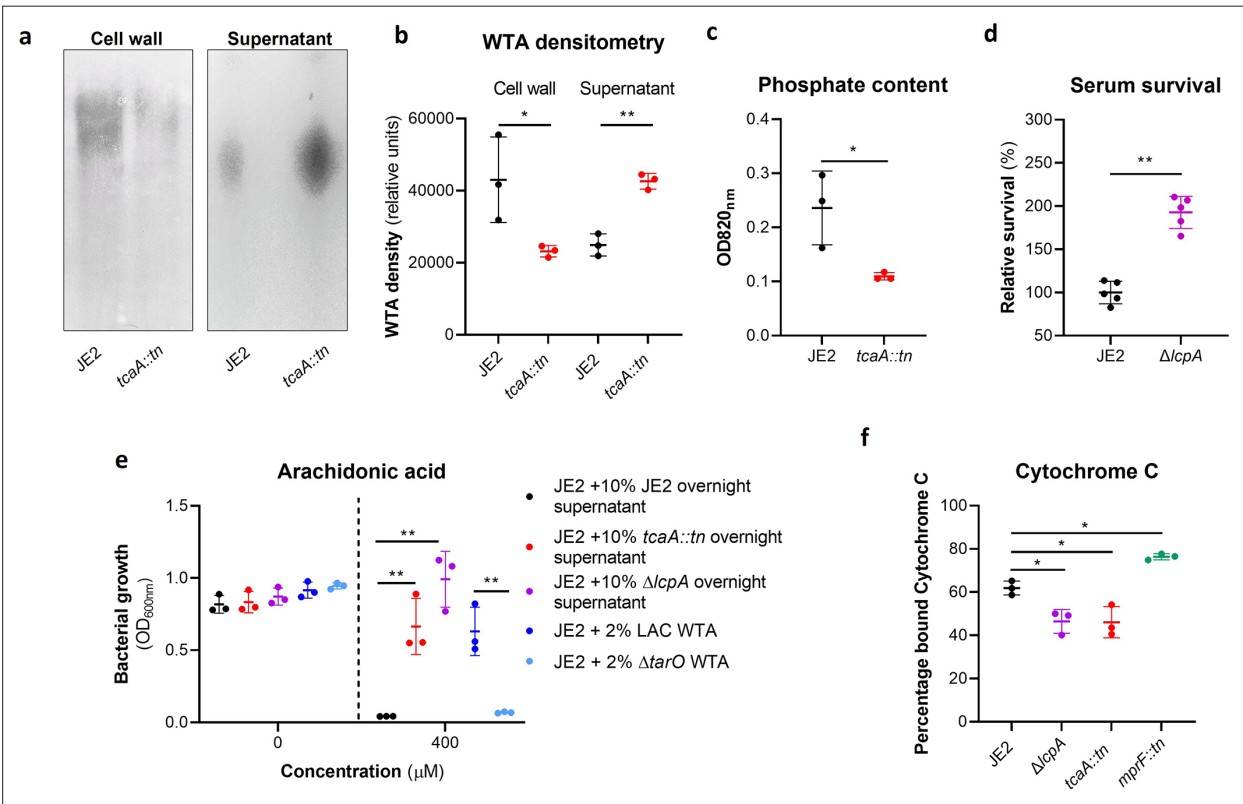

**Figure 5.** Wall teichoic acids (WTA) are released from the cell wall in the *tcaA* mutant to affect resistance to host defence fatty acids (HDFAs) and antimicrobial peptides (AMPs). (**a**) WTA was extracted from both cells and supernatant of the wild type and *tcaA* mutant of *S. aureus* and visualised on and SDS-PAGE gel stained with 1 mg/ml Alcian blue. The *tcaA* mutant had significantly less WTA in the cell wall but more in the supernatant. (**b**) The WTA extractions were performed on the wild type and tcaA mutant in triplicate the relative density of the WTA quantified by densitometry. (**c**) The phosphate content of the cell wall WTA extracts was quantified, which verified that the *tcaA* mutant has significantly less WTA. (**d**) The inactivation of the *lcpA* gene increases the sensitivity of *S. aureus* to killing by serum. (**e**) The wild type strain JE2 was grown in broth supplemented (at 10%) with supernatant of either itself (JE2), with that from a *tcaA* mutant, or with that from a *lcpA* mutant. These supernatants had no effect on the growth of JE2 in the absence of arachidonic acid. In the presence of arachidonic acid JE2 was unable to grow when supplemented with its own supernatant, however, when supplemented with the supernatant of the two mutants, which both contain soluble WTA, JE2 was able to grow. The addition of purified WTA extract (at 2%) from another *S. aureus* strain (LAC) also neutralised arachidonic acid, however an equivalent extract from an isogenic WTA mutant (LAC Δ*tarO*) did not. (**f**) The charge across the cell wall of the wild type and *tcaA* mutant was compared using cytochrome c, where the mutant was found to be less negatively charged. An *mprF* mutant has been included as a control. The dots represent individual data points (n=5 for (**d**), n=3 for remaining assays), the bars the mean value, and the error bars the standard deviation. Significance was determined as *<0.05, **<0.01, ***<0.001, ****<0.0001.

The online version of this article includes the following source data for figure 5:

**Source data 1.** Uncropped gel of wall teichoic acid (WTA) extracts from the cell wall of the wild type and *tcaA* mutant in triplicate.

**Source data 2.** Uncropped gel of wall teichoic acid (WTA) extracts from the supernatant of the wild type and *tcaA* mutant in triplicate.

present in the cell wall they are believed to be protective by interfering with the penetration of the hydrophobic HDFAs through to the bacterial membrane (*Kohler et al., 2009*). It has also been shown that when the ligation of WTA to the cell wall is affected, such that they are instead released into the environment, this also decreases the sensitivity of the bacteria to HDFAs (*Beavers et al., 2019*). With distinct modes of action and means of accessing their target, it is intriguing to consider how TcaA may be contributing to increasing the sensitivity of *S. aureus* to both these types of antibacterial molecules. To examine whether the TcaA-associated increase in sensitivity to arachidonic acid is a result of an increase in the abundance of WTA in the bacterial cell, we extracted and quantified WTA from the wild type and *tcaA* mutant, where we found that the cell wall of the wild type had significantly more WTA when compared to the *tcaA* mutant cell extract (*Figure 5a, b, and c* and *Figure 5—source data 1*). We next quantified WTA in the bacterial supernatant, as recent work on a *S. aureus lcpA* mutant demonstrated that another WTA-related means of altering sensitivity to arachidonic acid is to release the WTA to the bacteria surface (*Beavers et al., 2019*). We found significantly less WTA in the extract of the TcaA producing wild type strain compared to that of the *tcaA* mutant (*Figure 5a and b* and *Figure 5—source data 2*), which could explain the observed difference in sensitivity to HDFAs. It was unclear from the previous work on LcpA how the released WTA by the *lcpA* mutant affects arachidonic acid sensitivity, and here we demonstrate that this mutant, like the *tcaA* mutant, is also less sensitive to human serum (*Figure 5d*). One hypothesis is that the WTA can either sequester or inactivate HDFAs in the environment surrounding the bacteria, thereby neutralising them. To test this, we harvested the supernatant of the wild type JE2, the JE2 *tcaA* mutant, and the JE2 *lcpA* mutant from overnight growth. These supernatants were used to supplement fresh broth (at 10%) into which the wild type JE2 strain was inoculated. In the absence of any arachidonic acid JE2 grew equally well regardless of the supernatant supplement (*Figure 5e*). However, in the presence of the supernatant of both the *tcaA* and *lcpA* mutant (both of which contain an abundance of released WTA), JE2 was able to grow in the presence of arachidonic acid; whereas when the JE2 supernatant was used as the supplement the bacteria were unable to grow (*Figure 5e*). To further verify that soluble WTA can neutralise arachidonic we performed WTA extractions from an isogenic wild type and mutant of the *S. aureus* strain LAC where the mutant has had the *tarO* gene deleted and consequently does not produce any WTA (*Figure 5e*). The incorporation of the wild type LAC purified WTA extract to broth at 2% created an environment in which JE2 could grow in the presence of arachidonic acid, whereas the equivalent WTA extract of the *tarO* mutant, which contains no WTA, did not (*Figure 5e*). Together these data demonstrate that WTA in the environment can neutralise arachidonic acid, and this provides a likely explanation for the increased sensitivity to HDFAs associated with TcaA production.

### The reduced abundance of WTA in the bacterial cell wall contributes to the resistance of *S. aureus* to AMPs

In addition to being hydrophilic, WTA is also predominantly negatively charged (*Xia et al., 2010*). Given that a change in charge across the bacterial cell wall is frequently associated with resistance to AMPs (*Peschel et al., 1999*; *Ernst and Peschel, 2011*; *Joo and Otto, 2015*), and that the wild type strain has more WTA in its cell wall, this may explain the increased AMP sensitivity of the TcaA producing strain. To examine this, we incubated the bacteria with cytochrome *c*, which is positively charged and through its electrostatic-related ability binds to the bacterial cells, providing a measure of the charge across bacterial cell walls. Using this we found that the wild type strain bound significantly more cytochrome *c* and is therefore more negatively charged than the *tcaA* mutant, which explains the increase in AMP sensitivity we have found associated with TcaA production (*Figure 5e*). Together these results explain how TcaA production affects the sensitivity of *S. aureus* to both HDFAs and AMPs. By retaining WTA within the cell wall, there is less released to sequester or inactivate the fatty acids from the environment, and due to their negative charge, increased amounts of WTA in the cell wall affects the charge across the cell wall and results in increased electrostatic attraction of the positively charged AMPs.

### Serum-induced tcaA expression increases both the abundance of WTA in the cell wall and the sensitivity *S. aureus* to teicoplanin

As mentioned previously, an additional interesting feature of the TcaA protein is that it has also been associated with changing the sensitivity of the bacteria to the antibiotic teicoplanin, although this

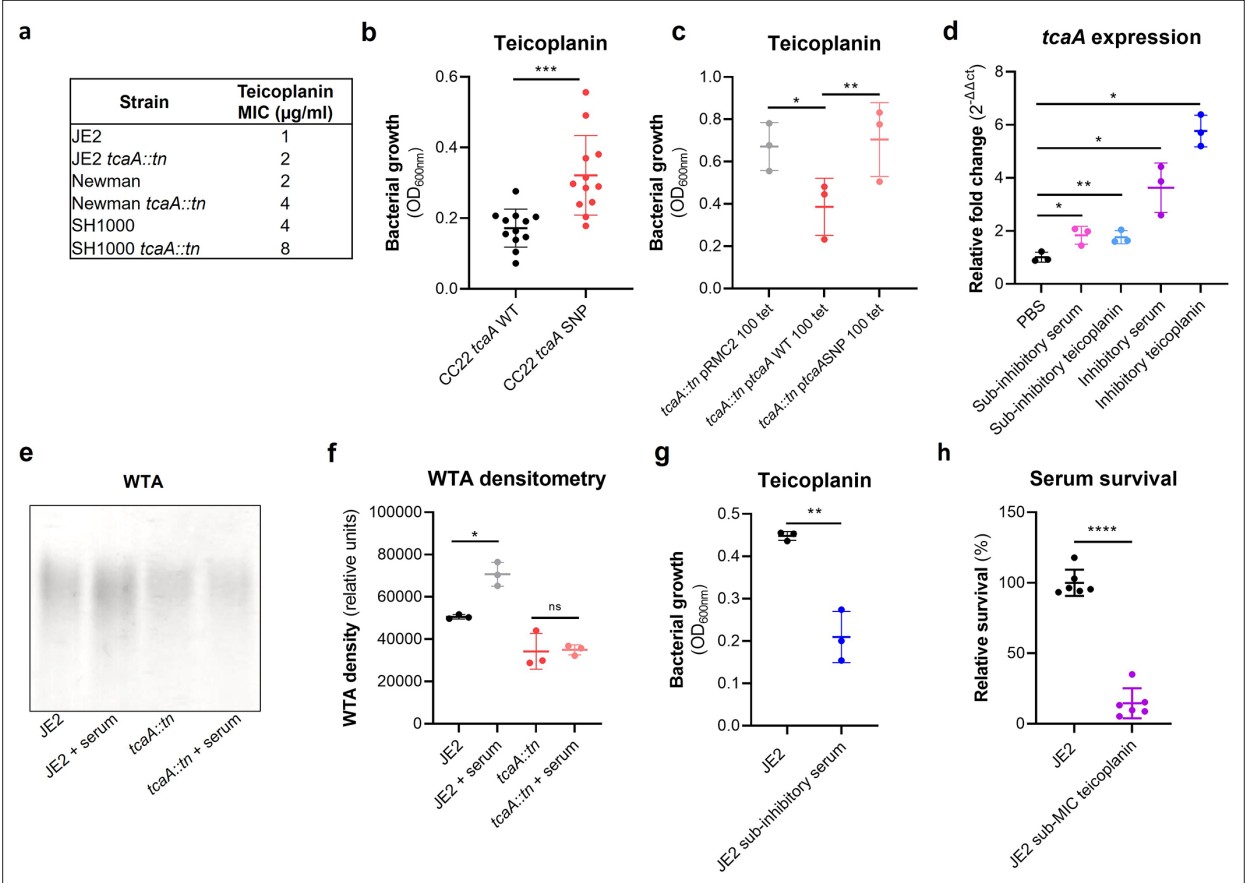

**Figure 6.** TcaA confers increased sensitivity to teicoplanin. (**a**) The inactivation of the *tcaA* gene increases the minimum inhibitory concentrations (MICs) in three *S. aureus* backgrounds. (**b**) The clinical isolates with polymorphisms in the *tcaA* gene were on average less sensitive to teicoplanin (0.5 µg/ ml) than those with the wild type gene. (**c**) The most common *tcaA* single nucleotide polymorphism (SNP) decreases the sensitivity of *S. aureus* to teicoplanin (0.5 µg/ml). (**d**) The expression of the *tcaA* gene was quantified by qRT-PCR in both subinhibitory and inhibitory concentrations of serum and teicoplanin. There was a dose-dependent effect on *tcaA* induction for all concentrations used. (**e**) Exposure of JE2 to inhibitory concentrations of human serum resulted in an increase in wall teichoic acid (WTA) abundance in the bacterial cell wall, an effect not seen when the *tcaA* mutant was exposed to serum. (**f**) The WTA extractions were performed on the wild type and tcaA mutant in triplicate the relative density of the WTA quantified by densitometry. (**g**) Growth of JE2 in the presence of subinhibitory concentrations of teicoplanin (0.5 µg/ml) over a 24 hr period was quantified following pre-exposure to subinhibitory concentrations of human serum (2.5%), where pre-exposure to serum increased sensitivity to teicoplanin. (**h**) The ability of *S. aureus* to survive exposure to subinhibitory concentrations of serum (2.5%) following overnight growth in a subinhibitory concentration of teicoplanin (0.5 µg/ml), where pre-exposure to teicoplanin increases the sensitivity to serum. The dots represent individual data points (n=12 for (**b**), n=3 for (**c**, **d**, **f**, **g**), n=6 for (**h**)) for the serum, the bars the mean value, and the error bars the standard deviation. Significance was determined as *<0.05, **<0.01, ***<0.001, ****<0.0001.

The online version of this article includes the following source data and figure supplement(s) for figure 6:

**Source data 1.** Uncropped gels of wall teichoic acid (WTA) extracts from cell wall extracts of a wild type and *tcaA* mutant incubated with and without human serum, performed in triplicate.

**Figure supplement 1.** Induction of expression of the *tcaA* gene by arachidonic acid and LL37.

has been reported to vary between strains (*Brandenberger et al., 2000*; *Maki et al., 2004*). To examine this for our strains we determined the minimum inhibitory concentrations (MICs) for three independent sets of isogenic wild type and tcaA mutants (i.e., in strains JE2, Newman, and SH1000, all corresponding to CC8), where TcaA production resulted in increased sensitivity to this antibiotic across all three strains (*Figure 6a*). To examine whether the clinical isolates with the SNPs in the *tcaA* gene associated with increased serum resistance also had altered teicoplanin sensitivity, we compared the ability of 12 clinical isolates containing the GWAS identified *tcaA* SNPs to grow in teicoplanin to that of 12 isolates from the same collection with the wild type gene sequence, that is, without the polymorphism. The isolates with the polymorphism in the *tcaA* gene were on average less sensitive to

teicoplanin (*Figure 6b*). Using the pRMC2 complementing plasmid expressing either the wild type or SNP (conferring the Phe290-Ser substitution) containing *tcaA* gene, we demonstrated that this mutation significantly reduces the ability of the protein to sensitise the bacteria to teicoplanin (*Figure 6c*), confirming that this mutation negatively affects the activity of the TcaA protein. As TcaA has been shown to be upregulated when exposed to teicoplanin (*Brandenberger et al., 2000*; *Maki et al., 2004*), we examined whether serum would also induce its expression. The wild type JE2 strain was exposed to either subinhibitory (2.5% serum and 0.5 μg/ml teicoplanin) or inhibitory (10% serum and 10 μg/ml teicoplanin) concentrations of either serum or teicoplanin for 20 min, total RNA was extracted and the transcription of the *tcaA* gene quantified by qRT-PCR. For both serum and teicoplanin, induction of expression of the *tcaA* gene was concentration dependent (*Figure 6d*). Given the association of WTA abundance in the cell wall with *tcaA* expression described above (*Figure 5a–c*) we next examined whether serum-induced expression of *tcaA* would also result in an increase in abundance of WTA in the bacterial cell wall. Both the wild type and *tcaA* mutant were exposed to inhibitory concentrations of human serum for 90 min and the WTA extracted from the cell walls, where serum exposure resulted in an increase in WTA for the wild type strain but not the *tcaA* mutant (*Figure 6e and f* and *Figure 6—source data 1*). To examine whether serum-induced increased expression of *tcaA* would affect sensitivity to teicoplanin, we exposed the wild type bacteria to a subinhibitory concentrations of serum (2.5%) and then incubated these overnight in a subinhibitory concentration of teicoplanin (0.5 μg/ml). Pre-exposure to serum-induced increased sensitivity to teicoplanin suggesting that induction of *tcaA* expression by the initial serum exposure was sufficient to induce cross-sensitivity to the antibiotic (*Figure 6g*). To put this sensitisation effect in the context of serum, the reverse experiment was also performed whereby the wild type strain was exposed to a subinhibitory concentration of teicoplanin (0.5 μg/ml) overnight and a serum killing assay performed following this incubation (*Figure 6h*). Pre-exposure to teicoplanin also induced increased sensitivity to serum again highlighting this cross-sensitivity effect. To verify this induction effect was not specific to the JE2 lineage, a *tcaA::gfp* reporter plasmid was constructed and electroporated into JE2, SH1000, Newman, and a clinical CC22 isolate EMRSA15. In each background serum, arachidonic acid and LL37 reduced *tcaA* expression with the exception of LL37 and the EMRSA15 strain (*Figure 6—figure supplement 1*). These findings suggest that upon entry into the bloodstream and exposure to many of the antibacterial components of serum, expression of the *tcaA* gene is induced and this sensitises *S. aureus* to killing by these components of serum.

## The inactivation of tcaA alters the antibiotic resistance profile of *S. aureus*

There is emerging evidence that WTA play a role in cell division and peptidoglycan biosynthesis (*Schlag et al., 2010*). In support of this, it has been reported that a *tarO* mutant, devoid of WTA, confers sensitisation to β-lactam antibiotics due to mislocalisation of the penicillin binding proteins (PBPs *Brown et al., 2012*). As the target for teicoplanin is peptidoglycan, more specifically the terminal D-Ala-D-Ala residue of lipid II, we hypothesised that the increased ligation of WTA into the cell wall of the TcaA producing strain may affect peptidoglycan biosynthesis and assembly, and that this may be responsible for the change in sensitivity to teicoplanin. To test this further, we compared the sensitivity of the wild type and mutant to a range of peptidoglycan attacking antibiotics, where we found the TcaA producing wild type strain to be more sensitive to vancomycin (as shown previously; *Maki et al., 2004*), oritavancin, but not dalbavancin (all alternative members of the glycopeptide class of antibiotics), and ramoplanin, a glycolipodepsipeptide antibiotic (*Figure 7—figure supplement 1*). However, TcaA production conferred decreased sensitivity to oxacillin, a β-lactam antibiotic that targets penicillin binding protein 2 and moenomycin a phosphoglycolipid antibiotic that inhibits the transglycosylase PBP enzymes and prevents the formation of peptidoglycan polymers (*Figure 7—figure supplement 1*). This suggests that TcaA production affects the overall composition and structure of peptidoglycan in the wild type strain. To examine whether the crosslinking of peptidoglycan is affected by TcaA production, we tested the susceptibility of the wild type and *tcaA* mutant to lysostaphin, which is a glycylglycine endopeptidase capable of cleaving the pentaglycine crosslinks of peptidoglycan (*Kurokawa et al., 2013*). We found that lysostaphin cleaved the cell wall of the wild type strain more slowly than for the *tcaA* mutant (*Figure 7a*). We next performed a Triton X-100-induced autolysis assay where the wild type exhibited a slower rate of autolysis compared to

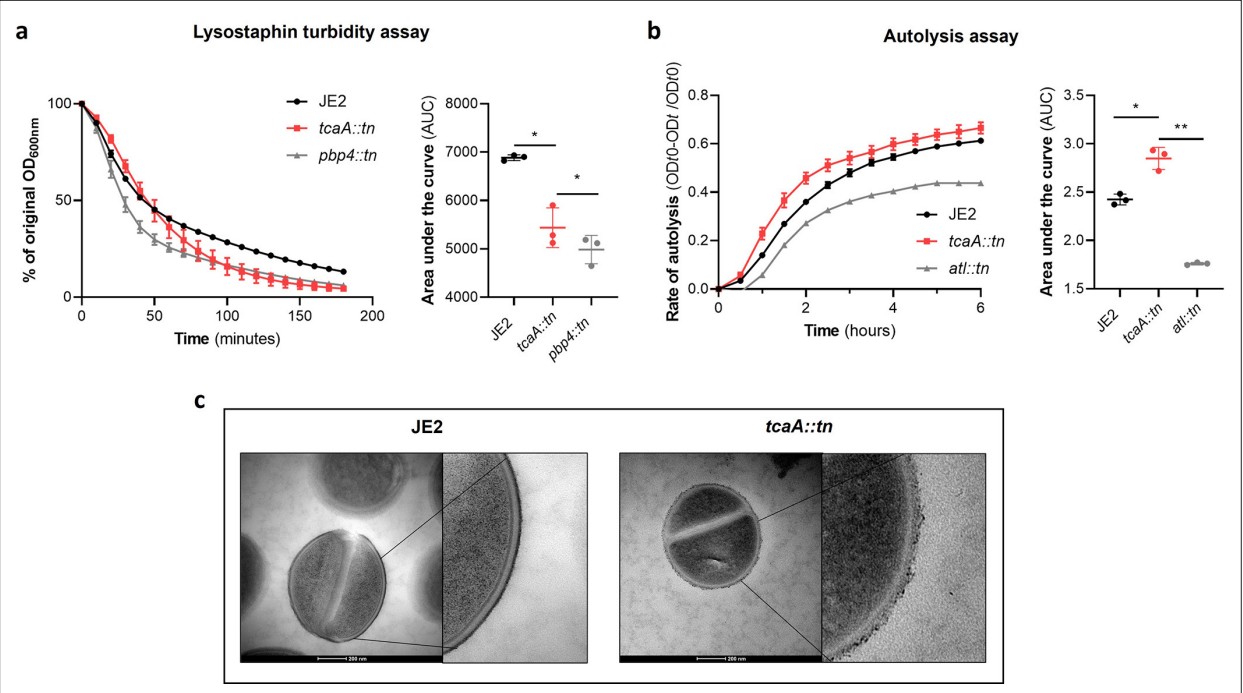

**Figure 7.** TcaA alters the structure of the *S. aureus* cell wall. (**a**) The rate of lysis in the presence of lysostaphin was assayed, where the strain producing TcaA was lysed at a slower rate than the *tcaA* mutant. A *pbp4* mutant was included as a control. Turbidity (OD$_{600}$) and area under the curves (AUCs) of the OD$_{600}$ data over time are presented. (**b**) The rate of autolysis of the bacteria in the presence of Triton X-100 was assayed where the TcaA producing strain lysed at a slower rate relative to the *tcaA* mutant. An autolysin (*atl*) mutant was included as a control. Autolysis data and AUCs of the autolysis data over time are presented. The dots represent individual data points (n=3), the bars the mean value, and the error bars the standard deviation. For both (**a**) and (**b**) significance was determined as *<0.05, **<0.01 following AUC analysis of three biological replicates. (**c**) Transmission electron micrograph (TEM) of a wild type JE2 and *tcaA* mutant cell at two magnifications showing the smooth and consistent density of the cell wall when TcaA is produced, compared to the rough and patchy density of the mutant cell wall.

The online version of this article includes the following figure supplement(s) for figure 7:

**Figure supplement 1.** Relative growth of the wild type JE2 and *tcaA* mutant in a range of concentrations of cell wall attacking antibiotics: vancomycin, ramoplanin, dalbavancin, oritavancin, oxacillin, and moenomycin.

**Figure supplement 2.** Replicate transmission electron micrograph (TEM) images of wild type JE2 and *tcaA* mutant cells showing the relative rough and patchy cell wall of the mutant.

the mutant, suggesting that the cell wall is more robust due to higher levels of crosslinking when TcaA is produced (*Figure 7b*). Together these data suggest that the integrity of the peptidoglycan, in particular its crosslinking, is increased when TcaA is produced. This provides a likely explanation for why the mutant is more resistant to teicoplanin, reduced crosslinking in the cell wall increases the number of off target D-ala D-ala which teicoplanin will bind to instead of reaching lipid II in the membrane and causing disruption to this. With changes to the quantity and composition of WTA and peptidoglycan associated with TcaA production, we sought to visualise the cell wall using transmission electron microscopy. We observed a clear difference in the density and structure of the outer layer of the bacterial cell wall between the wild type and *tcaA* mutant, confirming a role for TcaA in the structural integrity of the *S. aureus* cell wall (*Figure 7c* and *Figure 7—figure supplement 2*).

## Once established, TcaA contributes positively to the development of bacteraemia

Our data shows that expression of *tcaA* is induced upon exposure to serum, that this can sensitise the bacteria to serum killing but also increase the abundance of WTA in the cell envelope. From a pathogenicity perspective these activities would appear to be in conflict, with one making it less likely for a bacterium to cause bacteraemia (increased serum sensitivity) and the other making it more likely for a bacterium to cause bacteraemia (increased WTA in the cell envelope; *Wanner et al.,*

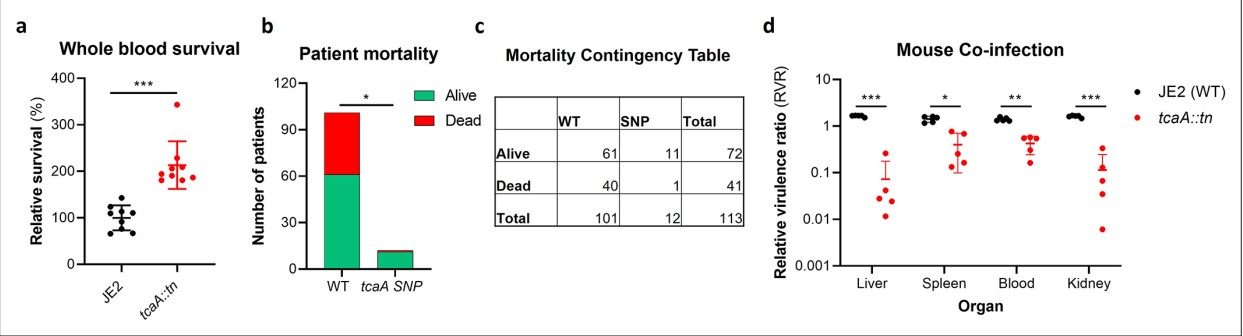

**Figure 8.** TcaA contributes to increased disease severity in mice and humans. (**a**) Whole human blood survival. The wild type JE2 and tcaA mutant were incubated in human blood from three donors (each in biological triplicate) for 90 min and their relative survival was quantified. TcaA production renders that bacteria more susceptible to killing. The dots represent individual data points (n=9), the bars the mean value, and the error bars the standard deviation. (b and c) A graph and contingency table showing data from 113 patient with *S. aureus* bacteraemia. The 30-day mortality rate was significantly higher for those infected with a *S. aureus* strain with a wild type *tcaA* gene compared to those with a mutated *tcaA* gene. (**d**) Mice were injected intravenously with 100 l of PBS containing an equal mixture of wild type and *tcaA* mutant ($2 \times 10^7$ colony forming units [CFUs] of each). After 48 hr, the ratio of the wild type and mutant were quantified, and the relative virulence ratio calculated. In the blood and all organs tested the wild type bacteria had a competitive advantage over the mutant demonstrating its increased relative virulence. The dots represent individual data points (n=5 per group), the bars the mean value, and the error bars the standard deviation. Significance was determined as *<0.05, **<0.01, and ***<0.001.

*2017*; *Weidenmaier et al., 2005*). To understand this apparent dichotomy we first examined whether TcaA production only affected survival in serum, whereas in whole blood where other components of the human immune system may be present (e.g., complement, antibodies, and phagocytes) this effect may be lost. Using human blood from three donors, we demonstrated that TcaA production increases the sensitivity of the bacteria to whole blood killing (*Figure 8a*). We next interrogated our human clinical data, where in addition to the 300 bacteraemia strains that form the basis of this study we have an additional 176 *S. aureus* isolates, which has allowed us to include more genetic diversity to this analysis (i.e., from CC22 [n=138], CC30 [n=162], ST8 [n=132], and ST93 [n=44]). The source of these isolates was also available to us (i.e., bacteraemia [n=341], SSTI [n=80], or carriage [n=55]), as well as their genome sequences (*Supplementary file 1*). We examined the distribution of nonsynonymous mutations in the *tcaA* gene across these, where it was present in 6% of the bacteraemia isolates compared with only 1.5% of non-bacteraemia isolates (n=21/341 and 2/135 respectively; p=0.03, in a Fisher's exact test). However, when we examined the 30-day mortality rates for the 113 bacteraemia patients for whom we had this clinical data, the mortality rate of the proportion of patients infected with a *S. aureus* strain with the wild type *tcaA* gene was significantly higher compared to those infected with a mutated *tcaA* gene (*Figure 8b and c*) (two-tailed chi-squared test: p=0.03). So, although the incidence of mutations in *tcaA* appears to be relatively rare, that isolates with variant *tcaA* genes are enriched amongst those causing bacteraemia suggests that a functional TcaA may limit the propensity of *S. aureus* to cause a bloodstream infection, such that it acts as a selective force for mutants of this gene. However, once the bacteraemia has become established, TcaA positively contributes to the infection process, likely due to the increased abundance of WTA in the cell envelope, and the greater structural integrity of the cell wall. To experimentally test these human findings we utilised a mouse model of bacteraemia where the tail vein of C57/Bl6 mice are inoculated with $2 \times 10^7$ colony forming units (CFUs) of bacteria to establish a bloodstream infection, where they seed from there into organs such as the liver spleen and kidneys. We performed mixed infections, where the wild type and mutant bacteria had to compete during the development of the infection within the mouse. The mice were infected with equal quantities of both bacteria and the change in ratio of the bacteria in the mouse blood, liver, spleen, and kidney recorded after 48 hr, which allowed the relative virulence ratio (RVR) of each bacterium to be calculated (*Monk et al., 2008*). The RVR of the TcaA producing wild type strain was significantly higher than that of the mutant at a systemic level (*Figure 8d*). As such, while the TcaA-associated self-sensitisation of the bacteria was not evident here, possibly due to the size of the inoculum level needed, once an infection has become established, TcaA contributes positively to disease severity as demonstrated by the wild type strain outcompeting the *tcaA* mutant.

# Discussion

To develop novel therapeutic approaches for infectious diseases, we need to understand how microorganisms cause disease and evade the host's immune system. To address this, we have applied a population-based approach to analyse the pathogenicity of *S. aureus* and identified seven genes that affect the bacteria's ability to survive exposure to human serum, the first step in the development of bacteraemia. The dissection of the molecular detail of how these genes affect serum survival is underway to determine their potential for therapeutic intervention. However, our focus here has been on the *tcaA* gene where we show that its expression is induced upon exposure to serum, that it is involved in the ligation or retention of WTA in the cell wall, and that this renders the bacteria more susceptible to killing by both AMPs and HDFAs. We have also found that TcaA is associated with changes to the crosslinking of peptidoglycan, such that the bacteria are more susceptible to some antibiotics (e.g., teicoplanin) and less susceptible to others (e.g., oxacillin). It is interesting to note that our GWAS approach identified *tcaA* due to the presence of function altering SNPs amongst clinical isolates. Our human data suggests that bacteraemia and/or

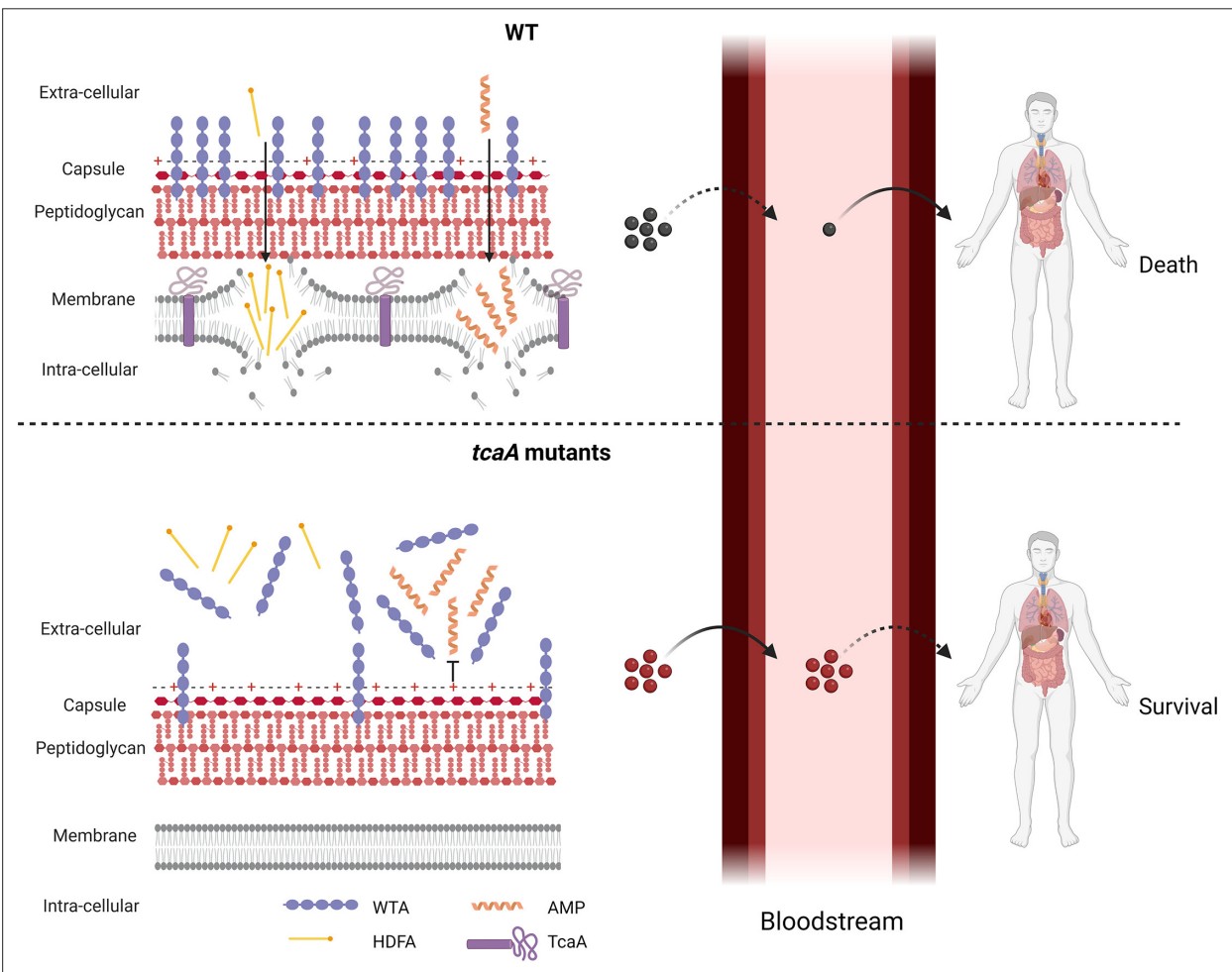

**Figure 9.** Graphical summary of our hypothesis for TcaA activity. Upon exposure to serum the expression of *tcaA* is induced in the wild type strain, resulting in an increased abundance of wall teichoic acid (WTA) in the cell wall and less in the extracellular milieu. This decreases the negative charge of the bacterial surface, rendering the bacteria more susceptible to antimicrobial peptides (AMPs). In addition, there is less WTA released into the extracellular milieu to sequester host defence fatty acids (HDFAs), rendering the bacteria more susceptible to killing by these molecules. This self-sensitising activity of TcaA limits the ability of *S. aureus* to establish a bloodstream infection. However, once established a strain with wild-type TcaA can cause more severe disease. In *tcaA* mutants, there is lower abundance of WTA in the cell wall and a higher abundance in the extracellular milieu. This increases bacterial resistance to AMPs as it increases the charge of the bacterial surface, resulting in less electrostatic attraction. The release of WTA into the extracellular milieu also sequesters HDFAs, preventing them from reaching the bacterial surface. As a result, a *tcaA* mutant is more likely to establish a bloodstream infection. However, this infection is less likely to cause severe disease due to the importance of TcaA in remodelling the cell wall for the full virulence of *S. aureus*. (Image created using BioRender.com.)

the antibiotics commonly used to treat it such as teicoplanin and vancomycin may impose a selective pressure on the bacteria to mutate this gene, due to the decreased sensitivity this confers to serum killing and to clinically relevant antibiotics. However, both our human and mouse data have confirmed that once bacteraemia has become established, TcaA contributes positively to disease progression, likely due to its cell wall remodelling activity. A graphical summary of this has been provided in *Figure 9*.

The *tcaA* gene is part of a three gene locus alongside the *tcaR* gene, which is predicted to encode a transcriptional regulator, and the *tcaB* gene which is predicted to encode an efflux pump (*Brandenberger et al., 2000*). This locus is also reported to be part of the *S. aureus* cell wall stimulon, following a number of studies that monitored the response of *S. aureus* to cell wall attack (*Steidl et al., 2008*; *Utaida et al., 2003*). It is interesting to note that the *tcaA* gene was not associated with serum resistance for the CC30 collection of isolates, however, there were very few isolates with variation in this gene amongst that collection such that they would have fallen below the minimum allele threshold. While work to understand the activity of each of the proteins encoded by the *tca* locus is underway, our data suggests that TcaA play a role in the ligation or retention of WTA within the cell wall, perhaps alongside the LcpA protein (*Beavers et al., 2019*). How this is achieved is yet to be determined. TcaA may play a direct role in ligating WTA to the cell wall; it may play a role in cell wall turnover and the subsequent release of WTA and other cell wall macromolecules into the external environment; or it may interact and subsequently interfere with other cell wall biosynthetic processes that are responsible for WTA ligation. It is interesting to note that *tarK*, which is one of the other six genes associated here with serum survival, is also involved in the biosynthesis of WTA. Additionally, *yfhO* which encodes an enzyme that glycosylates LTA is also involved in serum resistance (*Rismondo et al., 2018*), providing further evidence for the importance of teichoic acid-associated molecules to this aspect of the biology and pathogenicity of *S. aureus*. For TcaR, we hypothesise that it likely controls the transcription of the co-transcribed *tcaA* and *tcaB* genes, and may directly respond to the external stimulus of teicoplanin and serum as reported here. What is less clear is what role TcaB plays, as efflux pumps are more typically associated with increasing the resistance of a bacterium to an antibiotic, or other toxic molecules.

In addition to *tcaA* this study has identified one other locus (*yfhO*) that negatively affects serum survival, and added a further five loci (*tarK, gntR, ilvC, arsB*, and *pdhD*) to the list of genes that contribute positively to serum survival by protecting the bacteria from HDFAs and AMPs. This leads us to the questions of why *S. aureus* has loci in apparent conflict with regard to the survival of the bacteria during bacteraemia. It is perhaps through the consideration of the other ways in which *S. aureus* interact with humans that we may understand this. The vast majority of interactions that occur between *S. aureus* and humans is as a commensal in the human nose from which it can readily transmit. The five identified protective genes may therefore also protect the bacteria during colonisation, which would confer a selective advantage, and it is just an unfortunate coincidence that they also confer increased survival in serum during the development of bacteraemia. While the focus of this work has been on the role of TcaA in serum, it is also worth considering what role it may have during colonisation of the nose, which involves the adherence of the bacteria to the nasal epithelium, to which WTA has a well-established role. It is therefore possible that TcaA enhances the ability of *S. aureus* cells to colonise the nose by retaining WTA within the cell wall. However, the role of TcaA in responsiveness to cell wall attack is as yet unclear, but perhaps it contributes to the competitiveness of *S. aureus* within the nasal microbiome in response to cell wall damage inflicted by competitors.

The evolution of the virulence of a pathogen is heavily dependent upon their mode of transmission, which for opportunistic pathogens raises interesting challenges. If within-host invasiveness limits between-host fitness by preventing them from transmitting to a new host, there are likely selective pressures acting on bacterial populations to evolve means of limiting their ability to cause invasive disease. However, if increased invasiveness facilitates enhanced within-host fitness, as the bacteria multiply and spread throughout the body, the 'short-sighted evolution of virulence' hypothesis proposed by *Levin and Bull, 1994*, could apply and may explain situations like *S. aureus* bacteraemia. We propose that TcaA may represent a system at the cusp of both these scenarios, in that it responds to cell wall attack by serum by making itself more susceptible to killing by serum, thereby potentially limiting its ability to cause bacteraemia, which could have long-term between-host benefits. However, once established in the bloodstream and TcaA has remodelled the cell wall, the short-sighted view of

virulence hypothesis applies where the bacteria survive and thrive, allowing them to cause a successful bacteraemia.

Entry into the bloodstream to establish an infection has long been considered a significant bottleneck for the bacteria, but until now, the stringency of this bottleneck was believed to be a result of the onslaught of all the immune mechanisms present in blood. Here, we suggest that *S. aureus* may contribute to this bottleneck by responding to serum-induced cell wall damage by further sensitising itself to this attack as part of the pathway towards increasing their pathogenic capabilities by remodelling the cell wall. Our findings presented here opens up an entirely novel aspect to the biology of a major human pathogen and brings into question the appropriateness of the term 'opportunistic' when we refer to such highly effective bacterial pathogens.

## Materials and methods

### Bacterial strains and growth conditions

A list of the genetically amenable bacterial strains and mutants can be found in *Table 2*, and the clinical strains are listed in *Supplementary file 1*. All strains were cultured in tryptic soy broth (TSB) for 18 hr at 37°C with shaking. Nebraska transposon mutant library (NTML) (*Joo and Otto, 2015*) mutants were selected for using erythromycin (5 µg/ml). For the complemented pRMC2[45] strains, chloramphenicol (10 µg/ml) and anhydrous tetracycline (100 ng/ml) were added to the media where indicated.

### Serum and blood survival assay

Normal human serum was prepared from blood obtained from eight healthy volunteers using Serum CAT tubes (BD). On the day of collection, the blood was allowed to clot for 30 min at room temperature followed by 1 hr incubation on ice. Serum was extracted following two rounds of centrifugation at $700 \times g$ at 4°C for 10 min. Serum from individual donors were pooled, aliquoted, and immediately stored at –80°C, where each aliquot underwent only one round of freeze and thaw. The bacteria were grown overnight, and their density normalised to an $OD_{600nm}$ of 0.1 and 20 µl used to incubate in 180 µl 10% pooled human serum (diluted in PBS) for 90 min at 37°C with shaking. Serial dilutions were plated on tryptic soy agar (TSA) to determine CFUs. The same number of bacterial cells inoculated into PBS, diluted, and plated acted as a control. Survival was determined as the percentage of CFU in serum relative to the PBS control. Relative survival was determined through normalisation to JE2, which is presented as 100% in the graphs. The survival of each bacterial isolate was measured in triplicate and the mean of these data presented here.

### GWAS

Genome-wide association mapping was conducted using a generalised linear model, with serum survival as the quantitative response variable. We accounted for bacterial population substructure by adding to the regression model the first two component from a *principal component decomposition* of SNP data for each set of clinical samples (CC22 and CC30). The first two components accounted for 32% and 40% of the total variance for CC22 and CC30, respectively. In both cases, three distinct clusters were identified. We further considered a third model where we used cluster membership as covariates in our regression model, where clusters were defined using K-means clustering analysis (setting K=3); this, however, yielded identical results to the one based on PCA components. In total, 2066 (CC22) and 3189 (CC30) unique SNPs were analysed, the majority of which were subsequently filtered out for exhibiting a minor allele frequency of <0.03, reducing the data to 378 and 1124 SNPs, respectively. The p-values reported in *Table 1a and b* are not corrected for multiple comparison, however the Sidak method was used to correct for multiple comparisons and both significance thresholds are indicated in the Manhattan plots.

### Genetic manipulation of bacterial strains

Wild type genes were amplified by PCR from JE2 genomic DNA using the primers shown in *Table 3* and KAPA HiFi polymerase (Roche). The PCR product was cloned into the tetracycline-inducible plasmid pRMC2[26] using *Kpn*I and *Sac*I restriction sites and T4 DNA ligase (NEB). This was transformed into RN4220 and eventually into the respective NTML mutants through electroporation.

**Table 2.** Strains used in this study.

| Strain | Description | Reference |
|---|---|---|
| JE2 | USA300; CA-MRSA, type IV SCCmec; lacking plasmids p01 and p03; wild-type strain of the NTML | *Fey et al., 2013* |
| JE2 *tcaA:tn* | *tcaA* transposon mutant in JE2 | *Fey et al., 2013* |
| JE2 *tcaA:tn* pRMC2 | *tcaA* transposon mutant in JE2 transformed with empty pRMC2 vector | This study |
| JE2 *tcaA:tn* p*tcaA* | *tcaA* transposon mutant complemented with *tcaA* gene housed in pRMC2 expression plasmid | This study |
| JE2 *tarK:tn* | *tarK* transposon mutant in JE2 | *Fey et al., 2013* |
| JE2 *tarK:tn* pRMC2 | *tarK* transposon mutant in JE2 transformed with empty pRMC2 vector | This study |
| JE2 *tarK:tn* p*tarK* | *tarK* transposon mutant complemented with *tarK* gene housed in pRMC2 expression plasmid | This study |
| JE2 *gntR:tn* | *gntK* transposon mutant in JE2 | *Fey et al., 2013* |
| JE2 *gntR:tn* pRMC2 | *gntK* transposon mutant in JE2 transformed with empty pRMC2 vector | This study |
| JE2 p*gntR* | *gntK* transposon mutant complemented with *gntK* gene housed in pRMC2 expression plasmid | This study |
| JE2 *ilvC:tn* | *ilvC* transposon mutant in JE2 | *Fey et al., 2013* |
| JE2 *ilvC:tn* pRMC2 | *ilvC* transposon mutant in JE2 transformed with empty pRMC2 vector | This study |
| JE2 *ilvC:tn* p*ilvC* | *ilvC* transposon mutant complemented with *ilvC* gene housed in pRMC2 expression plasmid | This study |
| JE2 *arsB:tn* | *arsB* transposon mutant in JE2 | *Fey et al., 2013* |
| JE2 *arsB:tn* pRMC2 | *arsB* transposon mutant in JE2 transformed with empty pRMC2 vector | This study |
| JE2 *arsB:tn* p*arsB* | *arsB* transposon mutant complemented with *arsB* gene housed in pRMC2 expression plasmid | This study |
| JE2 *yfhO:tn* | *yfhO* transposon mutant in JE2 | *Fey et al., 2013* |
| JE2 *yfhO:tn* pRMC2 | *yfhO* transposon mutant in JE2 transformed with empty pRMC2 vector | This study |
| JE2 *yfhO:tn* p*yfhO* | *yfhO* transposon mutant complemented with *yfhO* gene housed in pRMC2 expression plasmid | This study |
| JE2 *pdhD:tn* | *pdhD* transposon mutant in JE2 | *Fey et al., 2013* |
| JE2 *pdhD:tn* pRMC2 | *pdhD* transposon mutant in JE2 transformed with empty pRMC2 vector | This study |
| JE2 *pdhD:tn* p*pdhD* | *pdhD* transposon mutant complemented with *pdhD* gene housed in pRMC2 expression plasmid | This study |
| RN4220 | Restriction deficient mutant of 8325–4 | *Kreiswirth et al., 1983* |
| DC10B | Δ*dcm* mutant in *Escherichia coli* DH10B (K12) | *Monk et al., 2012* |
| Mach1 | One Shot Mach1 T1 Phage-Resistant Chemically Competent *E. coli*. | Commercially available |
| SH1000 | MSSA, laboratory strain, 8325-4 with a repaired rsbU gene; SigB positive. | *Horsburgh et al., 2002* |
| SH1000 *tcaA::tn* | *tcaA* transposon mutant in SH1000 | This study |

*Table 2 continued on next page*

*Table 2 continued*

| Strain | Description | Reference |
|---|---|---|
| Newman | MSSA, laboratory strain isolated from human infection | *Hawiger et al., 1970* |
| Newman *tcaA*::tn | *tcaA* transposon mutant in Newman | This study |
| *tcaA*::tn p*tcaA*SNP | *tcaA* transposon mutant complemented with *tcaA* gene harbouring F290S SNP housed in pRMC2 expression plasmid | This study |
| *mprF*::tn | *mprF* transposon mutant in Newman | *Fey et al., 2013* |
| Δ*lcpA* | In frame unmarked deletion of *lcpA* generated by allelic exchange | *Beavers et al., 2019* |
| LAC | CA-MRSA USA300 type IV SCCmec | *Diep et al., 2006* |
| Δ*tarO* | *tarO* KO mutant in LAC | *Schuster et al., 2020* |
| JE2 pSB2019:*tcaA* | JE2 transformed with the pSB2019:*tcaA* reporter fusion | This study |
| SH1000 pSB2019:*tcaA* | SH1000 transformed with the pSB2019:*tcaA* reporter fusion | This study |
| Newman pSB2019:*tcaA* | SH1000 transformed with the pSB2019:*tcaA* reporter fusion | This study |
| EMRSA15 pSB2019:*tcaA* | SH1000 transformed with the pSB2019:*tcaA* reporter fusion | This study |

## Site-directed mutagenesis of the *tcaA* gene

Site-directed mutagenesis was used to mutate the wild type *tcaA* gene (Phe290) to the SNP *tcaA* (Ser290). Primers (5' ttttcaagttTcaaaacgtatggtc 3', 5' atacgttttgaAacttgaaaatgcc 3') were designed to have a complementary overlap of 21 nucleotides at the 5' end, with the nucleotide to be mutated at the centre of the overlap (the primers had non-complementary regions at the 3' end to facilitate annealing on the template). The p*tcaA* template was diluted to 33 ng/µl, and 1 µl was amplified using these primers and Phusion DNA polymerase. The PCR product was subsequently checked by agarose gel electrophoresis and was treated with 1 µl of DpnI directly in the PCR mix for 1 hr at 37°C. Then,

**Table 3.** PCR primers used in this study.

| Primer name | Primer sequence 5'–3' |
|---|---|
| *tcaA* FW | ataagcttgatggtaccaattgaagaggaacagaattgag |
| *tcaA* RV | tgaattcgagctcagatcatgacacatgcatcttatttaag |
| *tarK* FW | ataagcttgatggtacctttaataaaccgatgggggg |
| *tarK* RV | tgaattcgagctcagatgcaagtaatatatgccaacattag |
| *gntK* FW | ataagcttgatggtaccgtatgtcgtgtaatgaaggagtg |
| *gntK* RV | tgaattcgagctcagatttctttctttcattttaattcaacc |
| *ilvC* FW | ataagcttgatggtacccgaaacacaaaatatttaattatttgg |
| *ilvC* RV | tgaattcgagctcagattaacaactcctcattgtaggtctatc |
| *arsB2* FW | ataagcttgatggtaccgtcataaccttaagactggtgaatgc |
| *arsB2* RV | tgaattcgagctcagatgagttgcctgtacatataaaataaattg |
| *yfhO* FW | ataagcttgatggtacctgaaagttgaggataatgttgtg |
| *yfhO* RV | tgaattcgagctcagatgtagcaacattattttttgtcttgc |
| *pdhD* FW | ataagcttgatggtaccgaattattattaatggaggggtaaaac |
| *pdhD* RV | tgaattcgagctcagatgcatgtcacatgttaacgatatctc |

1 µl of the mixture was used to transform One Shot Mach1 T1 (Thermo Fisher) competent cells by heat shock then into RN4220 and finally *tcaA::tn* by electroporation.

## Phage transduction of the *bursa aurealis* transposon

The *tcaA bursa aurealis* transposon was phage transduced using the *S. aureus* phage Φ11 (*Krausz and Bose, 2016*). The phage lysate of the donor strain (*tcaA::tn*) was prepared as follows: The donor strain was grown overnight in TSB for 18 hr at 37°C with shaking. Two hundred microlitres of overnight culture was added to 25 ml of TSB containing 250 µl 1 M MgSO$_4$, 250 µl 1 M CaCl$_2$, 100 µl of Φ11 and left at 37°C with shaking until complete lysis was observed. The culture was subsequently pelleted at 12,000 × *g* for 3 min. The supernatant containing the lytic phage was filtered, sterilised, and stored at 4°C. Next, recipient strains (Newman and SH1000) were grown overnight in 20 ml of LK broth (1% Tryptone 0.5% Yeast Extract, 0.7% KCl) and pelleted at 2500 × *g* for 10 min. The supernatant was removed, and the pellets resuspended in 1 ml of fresh LK broth. Two reactions were then set up, one containing phage and one as a non-phage control. To the tube containing the phage, add 250 µl of the recipient strain (SH1000 or Newman), 250 µl of phage lysate, and 750 µl of LK broth containing 10 mM CaCl$_2$. To the tube containing the non-phage control, add 250 µl of the recipient strain (SH1000 or Newman) and 1 ml of LK broth containing 10 mM CaCl$_2$. Both reactions were incubated for 25 min at 37°C without shaking followed by 15 min at 37°C with shaking. Five hundred microlitres of ice-cold 20 mM sodium citrate was added followed by centrifugation at 10,000 × *g* for 10 min. Pellets were resuspended in 500 µl of 20 mM sodium citrate and incubated on ice for 2 hr. One hundred microlitres of culture were plated out on TSA containing 5 µg/ml erythromycin to select for the transposon. For the SH1000 phage transduction 100 µl were also plated out on oxacillin TSA plates as an added control to ensure there was no contamination of the phage lysate with JE2 *tcaA::tn*.

## Growth inhibition and MIC assays

To determine the relative inhibition of growth of the wild type and tcaA mutant by antibiotics, a broth microdilution method was used (*European Committee for Antimicrobial Susceptibility Testing (EUCAST) of the European Society of Clinical Microbiology and Infectious Diseases (ESCMID), 2003*). Briefly, overnight cultures were normalised to an OD$_{600nm}$ of 0.05 in cation-adjusted Mueller Hinton broth (MHB++) and 20 µl of resultant suspension used to inoculate 180 µl of fresh MHB++ containing a 1:2 dilution series of the respective antimicrobial agent. The ability of the bacteria to survive these was determined by quantifying bacterial growth (OD$_{600nm}$) using a CLARIOstar plate reader (BMG Labtech).

## AMP susceptibility

AMP susceptibility human neutrophil defensin-1 (hNP-1) (AnaSpec Incorporated, CA, USA) and LL-37 (Sigma) susceptibility assays were performed as described previously (*Duggan et al., 2020*). Briefly, overnight cultures were normalised to an OD$_{600nm}$ of 0.1 and incubated with 5 µg/ml of hNP-1 or LL-37 for 2 hr at 37°C. Serial dilutions were plated on TSA to determine CFUs. The same number of bacterial cells inoculated into PBS, diluted, and plated acted as a control. Survival was determined as the percentage of CFU on exposure to either LL-37 or HNP-1 relative to the PBS control. Relative survival was determined through normalisation to JE2.

## mRNA extraction

Overnight cultures were back diluted to an OD$_{600nm}$ of 0.05 in 50 ml of fresh TSB and grown to an OD$_{600nm}$ of 2. Two hundred microlitres of either PBS, 25% serum (final concentration 2.5%, subinhibitory), 5 µg/ml teicoplanin (final concentration 0.5 µg/ml, subinhibitory), 100% serum (final concentration 10%, inhibitory), or 100 µg/ml teicoplanin (final concentration 10 µg/ml, inhibitory) was added to 1.8 ml of bacterial culture and incubated for 20 min. RNA was extracted by Quick-RNA Fungal/Bacterial Miniprep Kit (Zymo Research) according to the manufacturer's instructions. RNA integrity was checked by running 5 µl aliquot of the RNA on a 1% agarose gel and observing the intensity of the ribosomal RNA. RNA samples were treated by TURBO DNase (Invitrogen) to eliminate any genomic DNA contamination. To verify that the samples were free from any DNA contamination, RNA samples were subjected to RT-qPCR alongside a no template control and 2.5 ng of a known JE2 genomic DNA and threshold rates compared.

## qRT-PCR

To quantify the expression of the *tcaA* RT-qPCR was performed using the housekeeping *gyrB* gene as a control. Complementary DNA (cDNA) was generated from mRNA using qScript cDNA Synthesis Kit (Quantabio). Following the manufacturer's protocol the cDNA was used as a template for the RT-qPCR. Primers used were as follows: *gyrB* FW 5' ggtgactgcattagatgtaaac 3', *gyrB* RV 5' ctgcttct aaaccttctaatacttgtatttg 3', *tcaA* FW 5' tagtttgcgcttcaggtg 3', *tcaA* RV 5' tgtggacataaatttgatagtcgtc 3'. The RT-qPCR was performed as follows: 10 μl 2× KAPA SYBR Mix, 1 μl of 10 μM forward primer, 1 μl of 10 μM reverse primer, 5 μl cDNA, and RNase-free water up to a total of 20 μl volume. The RT-qPCR was performed on a Mic qPCR cycler (biomolecular systems) and the cycling conditions consisted of an initial denaturation step of 95°C for 2 min, followed by 40 cycles of two-step cycling: 95°C 15 s, 60°C 1 min. RT-qPCR was carried out in technical triplicate for each sample and three biological repeats. The ratio of *tcaA* to *gyrB* transcript number was calculated using the $2^{-(\Delta\Delta Ct)}$ method (**Livak and Schmittgen, 2001**).

## WTA preparations

Crude WTA from murein sacculi was extracted for analysis by PAGE using adaptations of a previously described methodology (**Kho and Meredith, 2018**; **Brignoli et al., 2022**). Overnight cultures were washed once in buffer 1 (50 mM MES, pH 6.5) followed by centrifugation at $5000 \times g$. Cells were resuspended in buffer 2 (4% [wt/vol] SDS, 50 mM MES, pH 6.5) and boiled for 1 h. Sacculi were centrifuged at $5000 \times g$ and washed once in buffer 1, once in buffer 2, once in buffer 3 (2% NaCl, 50 mM MES, pH 6.5), once more in buffer 1 and finally resuspended in digestion buffer (20 mM Tris-HCl pH 8.0, 0.5% [wt/vol] SDS). To the digestion buffer suspension, 10 μl of proteinase K solution (2 mg/ml) was added and incubated on a heat block 50°C for 4 hr at 1400 rpm. Sacculi were centrifuged at $16,000 \times g$ and washed once in buffer 3, followed by three washes in $dH_2O$ to remove SDS. Sacculi were responded in 0.1 M NaOH and incubated for 16 hr at room temperature at 1400 rpm. Following the incubation, the sacculi were centrifuged at $16,000 \times g$, leaving the teichoic acids in the supernatant. Two hundred and fifty microlitres of 1 M Tris-HCL pH 6.8 was added to neutralise the NaOH and stored at –20°C.

WTA was also precipitated from the supernatant of overnight culture by adding three volumes of 95% ethanol and incubation at 4°C for 2 hr. Precipitated material was separated by centrifugation at $16,000 \times g$ for 15 min, washed once in 70% ethanol and resuspended in 100 mM Tris-HCl (pH 7.5) containing 5 mM $CaCl_2$, 25 mM $MgCl_2$, DNase (10 μg/ml), and RNase (50 μg/ml) and incubated for 3 hr at 37°C. The enzymes were heat inactivated at 95°C for 3 min. The supernatant WTA preparations were similarly stored at –20°C before being analysed by PAGE.

## WTA PAGE

WTA preparations were separated on tricine polyacrylamide gels using a Bio-Rad tetra cell according to a previously described method (**Brignoli et al., 2022**). The gels were separated at 4°C using a constant amperage of 40 mA under constant stirring until the dye front reached the bottom. Gels were washed three times in MilliQ $H_2O$ followed by staining in 1 mg/ml Alcian blue overnight. Gels were subsequently destained in in MilliQ $H_2O$, until the WTA became visible and finally imaged.

### Arachidonic acid supernatant conditioning

Overnight cultures of JE2, *tcaA*, and *lcpA* mutants were pelleted, and the supernatant saved. Growth inhibition assays were subsequently performed by adding 20 μl of $OD_{600nm}$ 0.1 bacterial suspension to 180 μl 10% overnight supernatant diluted in fresh MHB++ containing 400 μM of arachidonic acid (Sigma). Purified WTA extracts were also added to the growth media at a final concentration of 2%. The ability of the bacteria to survive the arachidonic acid was determined by quantifying bacterial growth ($OD_{600nm}$) following 24 hr at 37°C using a CLARIOstar plate reader (BMG Labtech).

### Cytochrome *c* binding assay

A cytochrome *c* binding assay was used to measure the relative surface charge of the bacteria (**Douglas et al., 2022**). Briefly, overnight cultures were normalised to an $OD_{600nm}$ of 8. The bacterial suspensions were washed twice in MOPS buffer (20 mM pH 7.0) and finally resuspended in 200 μl of MOPS buffer. Samples were then combined with 50 μl of cytochrome *c* (equine heart Sigma, 2.5 mg/ml in MOPS buffer) and incubated for 10 min at room temperature. Finally, samples were pelleted (16,000 × *g* for

1 min) and 200 µl of supernatant read for absorbance at $Abs_{530nm}$ using a SUNRISE Tecan microplate reader.

## Construction of a TcaA reporter fusion

A *tcaA* promoter-*gfp* fusion (pSB2019:*tcaA*) was constructed to determine expression of *tcaA* in response to several antimicrobial components of serum. A region of 289 bp upstream of the *tcaA* gene (**Dengler et al., 2011**) was amplified by PCR from JE2 genomic DNA using the primers *tcaA_promoter* FW 5'-atatgaattcagtattagaagtcatcaatca3' and *tcaA_promoter* RV 5'-atatccccgggtttcacctc aattctgttcct-3'. The pSB2019 vector was prepared by digesting pSB2031 (**Qazi et al., 2001**) from a previous study with EcoRI and SmaI to remove the RNAIII P3 promoter. The *tcaA* promoter region was also digested with EcoRI and SmaI and ligated into pSB2019 using T4 DNA ligase (NEB). This was transformed into DC10B (**Monk et al., 2012**), a strain of *E. coli* which mimics the adenine methylation profile of some *S. aureus* clonal complexes (including CC8 and CC22). The plasmid was then extracted and transformed into final strains via electroporation.

## TcaA induction assay

*S. aureus* strains carrying the pSB2019:*tcaA* plasmid were grown overnight in TSB with 10 µg/ml chloramphenicol. Each strain was then normalised to an $OD_{600nm}$ of 0.05 in fresh TSB and subcultured to an $OD_{600nm}$ of 0.5–0.6. Cultures were washed in PBS and concentrated to an $OD_{600nm}$ of 1 in PBS. One hundred microlitres of bacteria were combined with 100 µl of the appropriate antimicrobial compound (arachidonic acid or LL-37) in a black 96-well plate. GFP fluorescence ($485_{nm}$ excitation/$520_{nm}$ emission/1000 gain) was measured in a PHERAstar FSX plate reader (BMG Labtech) over 2.5 hr (readings every 30 min with 200 rpm shaking).

## Lysostaphin turbidity assay

Overnight cultures were back diluted to an $OD_{600nm}$ of 0.05 in 20 ml of fresh TSB and grown to an $OD_{600nm}$ 0.5–0.7. Cultures were normalised to give 1 ml of $OD_{600nm}$ 0.6. Lysostaphin was subsequently added to give a final concentration of 0.5 µg/ml. Two hundred microlitres of each strain were then added to a 96-well plate and incubated for 3 hr at 37°C in a CLARIOstar plate reader. $OD_{600nm}$ readings were taken every 10 min (500 rpm shaking before readings). Values were blank corrected according to 200 µl of PBS. The PBP4::*tn* mutant was included as a control.

## Triton X-100-induced autolysis assay

Overnight cultures were back diluted to an $OD_{600nm}$ of 0.05 in 20 ml of fresh TSB and grown to an $OD_{600nm}$ 0.5–0.7. Cells were washed once in ice-cold water (13,000 rpm 1 min) and resuspended in autolysis buffer (water containing 0.1% Triton X-100) to give an $OD_{600nm}$ of 1. 200 µl of each strain was then added to a 96-well plate alongside 200 µl of strains resuspended in water as a negative control and 200 µl of water as a blank. The *atl::tn* strain was used as a positive control. Strains were grown for 6 hr at 37°C in a CLARIOstar plate reader. $OD_{600nm}$ readings were taken every 30 min (500 rpm shaking before readings). The rate of autolysis was calculated as follows: (OD*t0*-OD*tn*/OD*t0*).

## Transmission electron microscopy

*S. aureus* strains JE2 and the TcaA mutant were pelleted in microfuge tubes and fixed by resuspending in 2.5% glutaraldehyde in cacodylate buffer (pH 7.3) kept at fridge temperatures until fixation was complete and then stored in the fridge. Pellets were resuspended in a BSA/glutaraldehyde gel at 10–20°C which sets after the cells were re-centrifuged into a pellet. The pellets were post-fixed with osmium ferrocyanide/osmium tetroxide mix in cacodylate buffer, en bloc stained with uranyl acetate (2%) and Walton's lead aspartate solutions, prior to dehydration in ethanol and infiltration with propylene oxide and Epon resin mix. Embedded blocks were polymerised for 24–48 hr at 60°C. Sections (70 nm) were cut on a Leica UC7 ultramicrotome and imaged using an FEI Tecnai T12 microscope.

## In vivo intravenous challenge model

For the mixed infection bacteraemia model, female, 7–8 weeks of age, C57BL/6NCrl mice (Charles River Laboratories) were used. The minimum numbers of animals required to obtain biological and

statistically significant effects to achieve our scientific objectives was determined to be 5. This numbers is based on our extensive prior experience in performing similar studies and through consultation with a biostatistician who provided input on study design and undertook power calculations. Each mouse was allocated randomly to a specific experimental group and experimental groups housed in the same cage. Bacteria were prepared as described previously (*Nguyen et al., 2022*).

Briefly, 1:100 dilutions of overnight bacterial cultures were grown to mid-log phase (~2 hr) in TSB with shaking at 180 rpm at 37°C. Bacteria were then harvested, washed with PBS, and adjusted spectrophotometrically at $OD_{600nm}$ to obtain the desired bacterial concentration of ~4.8 $10^8$ CFU/ml. A 50:50 mixture of JE2 and *tcaA::tn* were made and 100 l was injected intravenously via the tail vein. The final CFU for each strain per mouse was ~2.4 $10^7$ CFU. After 48 hr, terminal cardiac bleeds were performed on anaesthetised mice and whole blood was transferred into heparin tubes (Sarstedt) and mixed to prevent coagulation. Mice were then euthanised by $CO_2$ inhalation. Organ homogenates of spleens, livers, and kidneys were performed as previously described (*Bae et al., 2021*) and serial dilutions of each sample were plated onto TSA plates with and without erythromycin (5 µg/ml) for bacterial enumeration. The CFU of JE2 was calculated by subtracting CFU on TSA with erythromycin (*tcaA::tn* only) from CFU on TSA plates (JE2 and *tcaA::tn*).

## Statistics

Paired two-tailed Student's t-test or one-way ANOVA (GraphPad Prism v9.0) were used to analyse the observed differences between experimental results. A p-Value <0.05 was considered statistically significant and the Sidak method was used to correct for multiple comparisons.

## Materials availability statement

All genetically manipulated strains created in this study are available at the University of Bristol. Please contact the corresponding author to discuss and arrange this.

## Acknowledgements

We would like to thank Prof. Mark Jepson, Chris Neal, and Lorna Hogdson at the Wolfson Bioimaging Suite. We would also like to thank Eric Skaar and William Beavers for providing the *lcpA* mutant. Funding information: EJAD was funded on a studentship provided by the School of Cellular and Molecular medicine, University of Bristol. NP is funded on a BBSRC SWBio DTOP PhD Studentship. DA is funded by a PhD studentship awarded by the Saudi Arabian Cultural Bureau. RCMs and RMM are both Wellcome Trust Investigators (Grant reference numbers: 212258/Z/18/Z and 202846/Z/16/Z). MO is supported by the Intramural Research Program of the National Institutes of Allergic and Infectious Diseases (NIAID), US National Institutes of Health (NIH), project number ZIA AI000904. ML acknowledges funding from the Academy of Medical Sciences (SBF006\1023).

## Additional information

### Funding

| Funder | Grant reference number | Author |
| --- | --- | --- |
| University of Bristol | PhD Studentship | Edward JA Douglas |
| Biotechnology and Biological Sciences Research Council | SWBio DTP | Nathanael Palk |
| Saudi Arabian Cultural Bureau | | Dina Altwiley |
| Wellcome Trust | 212258/Z/18/Z | Ruth C Massey |
| Wellcome Trust | 202846/Z/16/Z | Rachel M McLoughlin |
| National Institute of Allergy and Infectious Diseases | ZIA AI000904 | Michael Otto |

| Funder | Grant reference number | Author |
|---|---|---|
| Academy of Medical Sciences | SBF006\1023 | Maisem Laabei |

The funders had no role in study design, data collection and interpretation, or the decision to submit the work for publication. For the purpose of Open Access, the authors have applied a CC BY public copyright license to any Author Accepted Manuscript version arising from this submission.

## Author contributions

Edward JA Douglas, Conceptualization, Data curation, Formal analysis, Validation, Investigation, Visualization, Methodology, Writing – original draft, Project administration, Writing – review and editing; Nathanael Palk, Conceptualization, Data curation, Formal analysis, Validation, Investigation, Methodology, Writing – original draft, Project administration, Writing – review and editing; Tarcisio Brignoli, Conceptualization, Formal analysis, Investigation, Methodology, Writing – review and editing; Dina Altwiley, Marcia Boura, Ryan Liu, Roger C Hsieh, Eoin O'Brien, Investigation, Methodology; Maisem Laabei, Resources, Supervision, Writing – review and editing; Mario Recker, Conceptualization, Data curation, Formal analysis, Investigation, Methodology, Writing – original draft, Writing – review and editing; Gordon YC Cheung, Conceptualization, Supervision, Investigation, Methodology; Michael Otto, Conceptualization, Resources, Supervision, Funding acquisition, Project administration, Writing – review and editing; Rachel M McLoughlin, Conceptualization, Resources, Formal analysis, Supervision, Funding acquisition, Investigation, Methodology, Writing – review and editing; Ruth C Massey, Conceptualization, Resources, Formal analysis, Supervision, Funding acquisition, Validation, Investigation, Visualization, Methodology, Writing – original draft, Project administration, Writing – review and editing

## Author ORCIDs

Edward JA Douglas ⓘ http://orcid.org/0000-0001-5892-8539
Nathanael Palk ⓘ http://orcid.org/0000-0002-6918-8852
Maisem Laabei ⓘ http://orcid.org/0000-0002-8425-3704
Mario Recker ⓘ http://orcid.org/0000-0001-9489-1315
Ruth C Massey ⓘ http://orcid.org/0000-0002-8154-4039

## Ethics

Human subjects: For the human serum collection, all healthy volunteers provided written informed consent and all methods and experimental protocols were carried out in accordance with the recommendations of the University of Bath, Research Ethics Approval Committee for Health. The present study was approved by the University of Bath, Research Ethics Approval Committee for Health [reference: EP 18/19 108].

C57/Bl6 mice were bred in-house in Trinity College Dublin and C57BL/6NCrl mice were purchased from Charles River Laboratories. Mice were housed under specific pathogen-free conditions at the Trinity College Dublin Comparative Medicines unit and maintained under pathogen-free conditions in an Association for Assessment and Accreditation of Laboratory Animal Care (AAALAC)-accredited animal facility in NIAID, NIH, respectively. All mice were female and used at 6-8 weeks. All animal experiments were conducted in accordance with the recommendations and guidelines of the health product regulatory authority (HPRA), the competent authority in Ireland and in accordance with protocols approved by Trinity College Dublin Animal Research Ethics Committee and according to the regulations of NIAID's Division of Intramural Research Animal Care and Use Committee (DIR ACUC), animal study proposal LB1E.

Reviewer #1 (Public Review): https://doi.org/10.7554/eLife.87026.3.sa1
Reviewer #2 (Public Review): https://doi.org/10.7554/eLife.87026.3.sa2
Reviewer #3 (Public Review): https://doi.org/10.7554/eLife.87026.3.sa3
Author Response: https://doi.org/10.7554/eLife.87026.3.sa4

## Additional files

### Supplementary files

• Supplementary file 1. Table of the clinical isolates used this study which includes the Strain Identification code (Strain ID); what clonal complex or sequence type the isolates correspond to; the type of infection they were isolated from; the patient outcome after 30 days was known; whether they have a single nucleotide polymorphism (SNP) in their *tcaA* gene; and their genome sequence accession numbers.

• MDAR checklist

### Data availability

All data is available within the publication apart from the genome sequence data which is freely available as described below.

The following previously published dataset was used:

| Author(s) | Year | Dataset title | Dataset URL | Database and Identifier |
|---|---|---|---|---|
| Harris et al, Harris SR, Feil EJ, Holden MT, Quail MA, Nickerson EK, Chantratita N, Gardete S, Tavares A, Day N, Lindsay JA, Edgeworth JD, de Lencastre H, Parkhill J, Peacock SJ, Bentley SD | 2010 | Evolution of MRSA during hospital transmission and intercontinental spread | https://www.sanger.ac.uk/resources/downloads/bacteria/staphylococcus-aureus.html | Wellcome Sanger Institute, bacteria/staphylococcus-aureus |

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
