## [Editor Report · eLife assessment]

This **important** study uses an innovative GWAS approach and targeted testing to highlight *S. aureus* genes that modify susceptibility to serum, serum-derived antimicrobial products, and commonly used antibiotics. These findings are significant in that they highlight evidence of evolution of virulence determinants in the setting of exposure to host stressors expected to be present during bacteremia and antibiotic therapy. **Compelling** results build on a foundation of work attributing loss-of-function mutations in tcaA to glycopeptide non-susceptibility.

---

## [Referee Report · Reviewer #1 (Public Review)]

In this manuscript by Douglas et al, the investigative team seeks to identify *Staphylococcus aureus* genes (and associated polymorphisms) that confer altered susceptibility to human serum, with the hypothesis that such genes might contribute to the propensity of a strain to cause bacteremia, invasive disease, and/or death. Using an innovative GWAS-like approach applied to a bank of over 300 well-characterized clinical *S. aureus* isolates, the authors discover SNPs in seven different staphylococcal genes that confer increased survival in the setting of serum exposure. The authors then mainly focus on one gene, tcaA, and illustrate a potential mechanism whereby modification of peptidoglycan structure and WTA display leads to altered susceptibility to serum, serum-derived antimicrobial compounds, and antibiotics. One particularly significant finding is that the identified tcaA SNP is significantly associated with patient mortality, in that patients infected with the SNP bearing isolate are less likely to die from infection. It is therefore hypothesized that this SNP represents an adaptive mutation that promotes serum survival while decreasing virulence and host mortality. In a murine model of infection, the strain bearing the WT allele of tcaA is significantly more virulent than the tcaA mutant, suggesting that the role of tcaA in bacteremia is infection-phase dependent.

This manuscript has many strengths. The triangulation of genomic analysis, patient outcomes data, and in vitro and in vivo mechanistic testing adds to the significance of the findings in terms of human disease. Testing the impact of mutating tcaA in multiple staphylococcal lineages and backgrounds also increases the rigor of the study. The identification of bacterial loci that impact susceptibility to both host antimicrobial compounds and commonly used antibiotics is also a strength of this work, given the evolutionary and treatment implications for such genes.

One moderate weakness is that the impact of the identified SNP in tcaA is only tested in some of the assays, whereas the majority of the testing is performed with a whole gene knockout. In some cases this results in more speculative conclusions that will require further testing to validate. All in all, this is an exciting manuscript that will be of interest to the broader research communities focused on staphylococcal pathogenesis, bacterial evolution, and host-pathogen interactions, as well as to clinicians who care for patients with invasive staphylococcal infection.

---

## [Referee Report · Reviewer #2 (Public Review)]

The authors embarked on a study to identify SNPs in clinical isolates of *S. aureus* that influence sensitivity to serum killing. Through a phenotypic screen of 300 previously sequenced *S. aureus* bacteremia (SAB) isolates, they identified ~40 SNPs causing altered serum survival. The remainder of the study focuses of tcaA, a gene with unknown function. They show that when tcaA is disrupted, it results in increased resistance to glycopeptides and antimicrobial components of human serum.

They perform an elegant series of experiments demonstrating how a tcaA knockout is more resistant to killing by whole serum. arachadonic acid, LL-37 and HNP-1. They provide compelling evidence that in the absence of tcaA resistance to arachidonic acid is mediated through release of wall teichoic acids from the cell wall, which acts as a decoy and sequesters the fatty acid.

Similarly, they suggest that resistance to cationic antimicrobial peptides is through alteration of the net charge of the cell wall due to loss of negatively charged WTAs based on reduced cytochrome C binding.

They continue to show that tcaA is induced in the presence of human serum, which causes increased resistance to the glycopeptide teichplanin.

They propose that tcaA disruption causes altered cell wall structure based on morphologic changes on TEM and increased sensitivity to lysostaphin and increased autolysis via triton x-100 assay.

Finally, they propose that tcaA influences mortality in SAB based on raw differences in 30-day morality. Interestingly they do decreased fitness during murine bacteremia model compared to wild-type.

The strengths of this manuscript are that it is well written and the identification of SNPs leading to altered serum killing is convincing and valuable data. The mechanism for tcaA-mediated resistance to arachadonic acid and AMPs is compelling and novel. The murine infection data demonstrating that tcaA mutants exhibit reduced virulence is important data.

The weakness of this manuscript mainly concerns the proposed mechanism that tcaA mutants show reduced peptidoglycan crosslinking. This conclusion is based on qualitative TEM images and increased sensitivity to lysostaphyin/autolysis. While these data are suggestive. it is difficult to draw such a conclusion without analysis of the cell wall by LC-MS.

Overall, I think this is a good submission and the majority of their conclusions are supported by the data. The mechanism behind the clinically relevant tcaA mutation is important, given its known role in glycopeptide resistance and therefore likely clinical outcomes. This manuscript would benefit from the inclusion of some additional experiments to help support their finding.

---

## [Referee Report · Reviewer #3 (Public Review)]

In this manuscript by Douglas et al., the authors used a functional genomics approach to understand how *Staphylococcus aureus* survives in the bloodstream to cause bacteraemia. They identified seven novel genes that affect serum survival. The study focused on tcaA, a gene associated with resistance to the antibiotic teicoplanin and is activated when exposed to serum and plays a role in producing a critical virulence factor called wall teichoic acids (WTA) in the cell envelope. This protein affects the bacteria's sensitivity to cell wall attacking agents, human defense fatty acids, and antibiotics, as well as autolytic activity and lysostaphin sensitivity. The data in this study suggested that TcaA play a role in the ligation or retention of WTA within the cell wall. However, more work is needed to clarify that part. Interestingly, despite making the bacteria more vulnerable to serum killing, tcaA contributes to *S. aureus* virulence by altering the cell wall architecture, as demonstrated by the wild type strain outcompeting the tcaA mutant in a Mouse Co-infection model. The study raises an important point that TcaA in *S. aureus* may represent a system balancing two scenarios: it makes the bacteria more susceptible to serum killing, potentially limiting bacteraemia and providing long-term benefits between hosts; however, once established in the bloodstream, the bacteria survive and thrive, causing successful bacteraemia, as per the short-sighted evolution of virulence hypothesis. This duality highlights the complex interplay between within-host and between-host fitness in bacterial evolution. I strongly suggest creating a graphical abstract to illustrate the complex relationship between within-host and between-host fitness scenarios involving TcaA. Having this visual representation in the discussion will enhance comprehension and provide a concise summary of the complex system for the reader.

In this manuscript, the authors achieved their aims, and the results support their conclusions. This work will be important for understanding this complex system and for developing novel therapeutics and vaccines for *S. aureus*.

---

## [Author Response]

The following is the authors' response to the original reviews.

We’d like to take this opportunity to thank the reviewers and editors for their consideration of our work. As detailed below, we have made the majority of the suggested corrections by the reviewers and believe these have greatly improved our manuscript. The reviewer’s comment are in blue font below and our response to each of these in black font.

**Reviewer #1 (Recommendations For The Authors):**
Suggestions to improve the manuscript:- Line 33 and 34: "This protein" is vague. Please reword to state whether you are referring to TcaA or to WTA

This has been corrected in the revised manuscript (Line 33)

- Intro: It would be helpful to provide more rationale for testing serum as a surrogate to whole blood in the GWAS screen. Serum is obviously lacking components of the clotting cascade, and some of these components have antimicrobial functions. However, this is easily justified in the text- e.g. to avoid clumping during the screen, to focus only on serum-derived antimicrobial compounds, etc.

This has been edited in the revised manuscript (Line 84-86)

- Line 120: Please state if the 300 clinical isolates represent 300 distinct patients, or if some of the isolates came from the same patient during sequential collections. If the latter, were there any instances in the which the tcaA SNP appeared during the course of infection?

They each came from individual patients so we were unfortunately unable to look for within host events. This information has been added to the revised manuscript (line 104).

- Line 133: the closed parenthesis sign is missing after "CC22"

This has been corrected in the revised manuscript (Line 135)

- Table 1a - NE1296 is misspelled as ME1296. Also there is a typo in the last entry of this table for the locus tag

This has been corrected in the revised manuscript.

- Table 1b - the authors should comment (in the discussion) on the potential reasons why tcaA was not identified in the CC30 background.

A comment to this effect has been added to the revised manuscript (Lines 553-59)

- Figure 2a - Why is the mutant with the empty complementation vector not significantly different from WT JE2?

The most widely used and reliable expression plasmid for complementation of mutated phenotypes in *S. aureus* is the pRMC2 plasmid, which requires chloramphenicol selection and anhydrotetracycline to induce expression of the cloned gene. These antibiotics, and the presence of the plasmid often affect the expression of other genes by the bacteria (as noted by this reviewer). As such, to verify complementation of a mutation the comparison we make is between the strain containing the empty plasmid induced with anhydrotetracycline with a strain with the gene containing plasmid induced with anhydrotetracycline. In that situation, the only difference between those two strains under those conditions is whether the gene is expressed or not. A comment explaining this has been added to the revised manuscript (lines 149-153).

- Line 188: Statistical analyses should be applied to figure 3C, which also appears to be underpowered.

P values have been added to this in the revised manuscript. We present data point of three biological replicates, which are the mean of three technical replicates, which we believe is sufficiently powers for this analysis.

- Figure 3 legend - Tecioplanin is mentioned in the title, but the data are not included here

This legend title has been the revised (Line 193).

- Figure 4 - here is an example where testing the actual tcaA SNP could have been enlightening. For example, what if the selective pressure makes the SNP more relevant to a specific AMP or AA?

While we agree that this would be an interesting experiment to perform, the complementing vector that we would need to use to compare the wild type and SNP contains gene requires antibiotics to select for the plasmid and another to induce expression. As such it becomes quite a complex and messy experiment where synergy between the antimicrobial agents would be likely, the results of which will be difficult to interpret.

- Lines 317-321 - Suggest moving this to discussion

We have left this here as we felt it a necessary summation/explanation of the results described in that section. It is discussed again later in the discussion section.

- Line 341 - I believe "serum" should actually be "teicoplanin"

This has been corrected in the revised manuscript (Line 342).

- Figure 6e - wouldn't it be more powerful to determine the WTA levels in the supernatants of these strains and conditions?

We could have done this both ways, but we focussed here only on how TcaA ligates WTA into the cell wall in the presence of serum.

- Figure 6 - What is the explanation for the different growth yields for JE2 in tecioplanin in panel A versus panel F? Are these actually two different concentrations? If so, please update the figure legend and the methods.

The concentration used for the A was inhibitory and for F sub-inhibitory. To improve the clarity of this we have now used a table displaying the MICs for the six strains as panel A. We have also included the concentration of teicoplanin used for each experiment in the legend.

- Line 413: Consider more precise language than "the cell wall is stronger". E.g. More crosslinks?

This has been edited in the revised manuscript (Line 421)

- Line 415: Consider changing "altered" to a directional term such as increases. It can be difficult for the reader to follow the expected change when you are discussing how the lack of a gene versus the presence of a gene changes susceptibility in one direction and another phenotype in the opposite direction.

This has been edited in the revised manuscript (Line 423).

- Figure 7: The conclusions made from panels A and B need to be supported by statistical analyses. It is unclear if these lines are truly different from one another.

These have been included in the revised fig 7.

- Line 426: I believe "tcaA" is missing following "producing"

This has been corrected in the revised manuscript (Line 434).

- Line 446: "increase" to "increases"

This has been corrected in the revised manuscript (Line 460).

- Figure 8C: if one goal of the mouse experiment was to look at survival during transit in whole blood, earlier timepoints are indicated based on the described kinetics of bloodstream dissemination in this model.

The primary goal of this experiment was to see if TcaA contributed positively or negatively to the development of the infection. Work on this protein is ongoing, and so we hope in coming years to be able to provide more detail on its activity in vivo.

- Line 506: "changes to the structural integrity of peptidoglycan" seems overstated without additional studies.

This has been edited in the revised manuscript (Line 524).

- Line 564: "represents" to "represent"

This has been corrected in the revised manuscript (Line 603).

- Line 588: The figures all refer to "100 net". Please confirm the concentration used.

This has been corrected in the revised manuscript (Line 628).

- Line 609: This refers to capsule production? Is this a copy error from a prior paper?

Yes it is, and has been corrected in the revised manuscript (Line 650).

- Line 763: Please provide the concentrations of arachidonic acid used for each experiment.

This has been included in the revised manuscript (Line 805)

- Line 836 and 837: This mentions a time course for blood culture from the infected mice. Where are these data?

Apologies, this is another cut and paste mistake from another paper, and had been removed.

- Line 870: please discuss how multiple comparisons testing was handled.

This has been included in the revised manuscript (Line 908).

- Supplemental figure 5 - Please add statistical analyses to support the conclusions in the manuscript. For example, there appears to be no differences for dalbavancin. Please also italicize tcaA in the legend.

These have been included and corrected in the revised manuscript.

**Reviewer #2 (Recommendations For The Authors):**
Line 65 - I would suggest adding the reference (doi: 10.1128/Spectrum.00116-21), which shows increased mortality in *S. aureus* bacteremia patients due to agr deficient isolates.

The suggested manuscript shows this effect of Agr dysfunction to be limited to patients with moderate to severe SOFA scores. As such it would require a nuanced description here that we think will detract from the flow of the introduction.

Line 68 - Please add DOI: 10.1016/j.cmi.2022.03.015 as a reference to support the mortality rate in *S. aureus* bacteremia. A systematic review and meta-analysis provides the highest level of evidence, and this is a contemporary study performed in 2022

This has been included in the revised manuscript (Line 68).

Line 70 - please add supporting reference for this statement

This has been included in the revised manuscript (Line 70).

Figure 2 - This image is low quality and appears pixelated. Please revise

This has been replaced with a higher resolution image in the revised manuscript.

Figure 3c Also appears slightly pixelated

This has been replaced with a higher resolution image in the revised manuscript.

Line 173 - I think it would helpful to mention the catalytic activity encoded by tcaA (aside from mediating sensitivity to glycopeptides) is unknown.

This has been included in the revised manuscript (Line 174)

Line 174 - also confers sensitivity to vancomycin https://doi.org/10.1128/AAC.48.6.1953- 1959.2004

This has been included in the revised manuscript, albeit at a later point than suggested here (Line 406)

Line 209 - did the authors test any other antimicrobial fatty acids such as palmitoleic acid? If common mechanism would also expect decreased sensitivity to other HDFA

No, we focused on arachidonic acid as this is the most relevant antimicrobial fatty acid in serum and it is produced by neutrophils and macrophages during the inflammatory burst.

Figure 4a-D: it would be useful to know what the MIC to these different components is and how that MIC relates to the concentration in human serum

We do not have MICs for all of these compounds tested here but can confirm that the concentrations used are physiologically relevant.

Figure 4b - Can you mention in the legend how the killing assays varied for arachadonic acid versus the other AMPs? I am not immediately clear how this experiment was performed, despite referring to methods

This has been included in the text of revised manuscript (Line 211-213) and the figure legend.

Figure 5 - there is no panel D

This has been corrected in the revised manuscript.

Figure 6a: Lines 328-329 state the experiment was performed in the MIC for each strain. The legend (line 374) states 0.5 ug/ml teicoplanin was used, which is below the MIC for all of the strains tested per supp table 2. Please correct this discrepancy.

This figure has been revised and the additional information included to improve the clarity of this section in the revised manuscript.

Figure 6a: On line 328, the authors state that the tcpA knockout increases the MIC for teicoplanin in each background. Figure 6a is performed in the presence of teicoplanin at 1x the MIC of the wild type (which will be below the MIC for the knockout). Therefore, we know each tcpA mutant will be able to grow in the presence of sub-mic concentrations of teicoplanin. Would a more informative way of conveying this information be to have MIC on the Y axis and background on the X axis?

This has been corrected and clarified in the revised manuscript with a table showing the MICs (fig. 6a).

Figure 6b-c: Similarly, would it be more helpful to show how the MIC varies with the different clinical isolate tcpA mutants?

While MICs have uses in clinical setting, they are a relatively crude and binary (growth V no growth) way to measure and compare sensitivity. For these two groups of isolates the MICs did not vary, which is why we used a concentration that sat that the threshold and quantified growth of all the isolates in this. This information has been added to the legend.

Figure 6e: The figure legends instructs us to refer to supplemental figure 3 to see the densiometry results. However, Figure 6e appears to be 4 conditions (WT and mutant +/- serum) and only examines the cell wall, whereas the supplemental figure refers to two conditions (WT + mutant) and looks at the cell wall and supernatant. I would recommend providing the densitometry data associated with the conditions in figure 6e, especially as differences seem more subtle by eye.

This has been included in the revised manuscript (fig. 6f)

Line 689-691 - description of teicoplanin concentrations used in figure 2. However, no teicoplanin was used in figure 2. Assume is referring to a different figure (figure 6?)

This has been corrected and clarified in the revised manuscript. Line 724.

Please add a section in the methods describing how the MIC was determined for JE2, SH1000 and Newman. Was it performed in CA-MHB or the media that the experiment in figure 6a was performed in. Serum can alter the MIC of several antibiotics

This has been corrected and clarified in the revised manuscript. Line 724-29.

Please add a section to the methods describing the whole blood killing assay, ideally describing how the blood was not frozen and used same day as venipuncture. This is important as freeze/thaw or time periods >12 hours are likely to severely effect the function of phagocytes, especially neutrophils.

This has been corrected and clarified in the revised manuscript. Lines 635-639

Line 588: ng/ul should read ng/µl

This has been corrected in the revised manuscript too ng/ml. Line 628

**Reviewer #3 (Recommendations For The Authors):**

We have now included a graphical abstract (Fig. 9)

Major:1- Line 102: I was not able to find the accession numbers of these 300 genomes, did the authors submit it to any public repository (e.g. NCBI)?

These were submitted previously to a public repository and the associated reference cited, but we have provided these in supplementary Table 1.

Minor:1 - Typo in line 133. Fix parenthesis after CC22.

Corrected.

2 - Typo: Fix figure 5 panels (5e should be 5d).

Corrected.

3 - Line 276: It is not clear why the extract for this experiment was supplemented at 2% while the other part of the experiment was done with 10%. Clarification is needed.

The experiments at 10% was using overnight supernatant, whereas those with 2% was a purified WTA extract. This has been clarified in the revised manuscript (lines 283 and in the figure legend)

4 - Line 278: Typo: Figure 6e should be figure 5d.

Corrected. (Line 278)

5 - Figure 5f: There is no explanation in the text or in the figure legend what the purpose of using mprF was.

A comment has been included in the figure legend.

6 - Line 328: It would be good if we the authors reports the CC of Newman and SH1000 for a better context for the readers.

This has been added. (Line 332)

7 - Line 341: Did the authors mean less sensitive to teicoplanin?

Corrected. (Line 342)

8 - Line 367: Dose dependent effect does not seem to be followed not only in panel H of Supp. Fig. 4(LL37 and EMRDA15) but also panels C, D and G.

Corrected.

9 - Line 587: Typo: Table 2.

These have all been corrected and/or clarified in the revised manuscript.